# Human commensal gut Proteobacteria withstand type VI secretion attacks through immunity protein-independent mechanisms

Nicolas Flaugnatti [1,5], Sandrine Isaac [1,5], Leonardo F. Lemos Rocha[1], Sandrine Stutzmann[1], Olaya Rendueles [2], Candice Stoudmann[1], Nina Vesel [1], Marc Garcia-Garcera [3], Amandine Buffet[2], Thibault G. Sana [1,4], Eduardo P. C. Rocha [2] & Melanie Blokesch [1✉]

While the major virulence factors for *Vibrio cholerae*, the cause of the devastating diarrheal disease cholera, have been extensively studied, the initial intestinal colonization of the bacterium is not well understood because non-human adult animals are refractory to its colonization. Recent studies suggest the involvement of an interbacterial killing device known as the type VI secretion system (T6SS). Here, we tested the T6SS-dependent interaction of *V. cholerae* with a selection of human gut commensal isolates. We show that the pathogen efficiently depleted representative genera of the Proteobacteria in vitro, while members of the *Enterobacter cloacae* complex and several *Klebsiella* species remained unaffected. We demonstrate that this resistance against T6SS assaults was mediated by the production of superior T6SS machinery or a barrier exerted by group I capsules. Collectively, our data provide new insights into immunity protein-independent T6SS resistance employed by the human microbiota and colonization resistance in general.

[1] Laboratory of Molecular Microbiology, Global Health Institute, School of Life Sciences, Ecole Polytechnique Fédérale de Lausanne (EPFL), Lausanne, Switzerland. [2] Microbial Evolutionary Genomics, Institut Pasteur, CNRS, UMR3525, 75015 Paris, France. [3] Department of Fundamental Microbiology, University of Lausanne, Lausanne, Switzerland. [4] Present address: LIPME, Université de Toulouse, INRAE, CNRS, 31320 Castanet-Tolosan, France. [5] These authors contributed equally: Nicolas Flaugnatti, Sandrine Isaac. ✉email: melanie.blokesch@epfl.ch

The human pathogen *Vibrio cholerae* is the causative agent of the severe diarrheal disease cholera, but its notoriously poor colonization ability of non-human adult animals makes it difficult to study. In fact, in one of the earliest studies on intestinal microbes, Metchnikoff suggested that adult experimental animals were refractory to the disease cholera due to the presence of their intestinal bacteria[1]. As a result, researchers have developed infant animal models (mice and rabbits) to study the pathogen's virulence potential, since infant animals lack a mature microbiota[1,2]. We now know that the disease cholera progresses first through toxin-coregulated pilus (TCP)-induced self-aggregation and microcolony formation in the gut, followed by the secretion of cholera toxin, which induces profuse diarrhea[3,4]. While useful for pathogenesis studies, infant animal models do not undergo the first step of intestinal colonization, which consists of the interaction of ingested *V. cholerae* with the mature microbiota. Because numerous studies have also shown that commensal microbes are critical in providing colonization resistance against incoming bacteria[5] and ultimately play a role in protecting humans from pathogens, it is important to study the interaction of *V. cholerae* with the human microbiota to better mimic the real-world infection conditions.

Intestinal pathogens can directly interact with the gut microbiota using strategies ranging from nutritional competition up to interbacterial warfare, with the latter encompassing the production of inhibitory molecules or contact-dependent inhibition/killing systems[6,7]. One example of a contact-dependent killing device is the type VI secretion system (T6SS), which was first described by Pukatzki and colleagues in 2006[8]. The presence of T6SS is widespread, as it is encoded by 25% of all sequenced Gram-negative bacteria[9], and more than 50% of ß- and γ-proteobacterial genomes harbor such a system[10]. The T6SS can be compared to an inverted contractile phage tail anchored to the cell envelope by a membrane complex[11]. A membrane complex-attached baseplate-like structure allows the polymerization of an internal tube made of Hcp protein rings, which is wrapped in a contractile sheath[11]. When the T6SS sheath contracts, the inner tube, the spike protein(s), and a cocktail of mostly tip-associated toxins is propelled into neighboring cells, causing growth inhibition or death[12]. The T6SS is therefore a well-suited nano-machine to drive interbacterial competition in the gut, as (i) the high bacterial density within this niche fosters direct contact between microbes, and (ii) the contact dependency of the T6SS limits collateral damage on non-neighboring bacteria.

There is evidence that the T6SS is important for colonization, as some intestinal pathogens, such as *Salmonella enterica* serovar Typhimurium, *Shigella sonnei*, and *V. cholerae*, are thought to utilize their T6SS to clear the resident microbiota and thereby promote their own colonization[13–15]. For instance, using the infant mouse cholera model, a recent study showed that *V. cholerae* outcompeted artificially pre-introduced mouse commensal *Escherichia coli* in a T6SS-dependent manner. T6SS-defective *V. cholerae* were therefore less abundant post-infection compared to their T6SS-positive parental strain[15]. This colonization defect was not observed when WT and T6SS-defective strains were co-administered, suggesting a global impact on niche clearing under the tested conditions.

Since the T6SS is an effective killing device, mechanisms need to exist to protect T6SS-positive bacteria and their siblings from (auto)intoxication. T6SS-positive bacteria therefore produce immunity proteins that directly interact with the cognate effector proteins and inhibit their toxic activity[16,17]. However, recent studies have also identified immunity protein-independent protection mechanisms[18]. For instance, the edited peptidoglycan of *Acinetobacter baumannii* provides protection from T6SS assaults[19]. Secreted exopolysaccharide (EPS), which is a primary component of bacterial biofilm matrices[20], was also shown to confer partial protection against external T6SS attacks, especially in quorum-sensing-impaired and therefore EPS-overproducing *V. cholerae*[21]. Moreover, Hersch and colleagues recently demonstrated that T6SS intoxication can activate protective envelope stress response (ESR) pathways such as the 'wall integrity gauge' system (WigKR[22]) in *V. cholerae* or the 'regulator of capsule synthesis' (Rcs) system in *E. coli* K-12[23]. Notably, despite its name, the Rcs system does not trigger the production of a membrane-tethered capsule in *E. coli* K-12, as the bacterium lacks amongst others the gene encoding the outer membrane tethering protein Wzi[24]. As a result, the synthesized polysaccharide (colanic acid) is secreted into the extracellular milieu and forms a biofilm-like structure referred to as slime[25]. Hence, the role, if any, of bona fide membrane-attached capsules in T6SS defense has not been investigated yet.

Here, we studied the impact of *V. cholerae*'s T6SS on human gut commensal Proteobacteria isolates. Indeed, previous studies had suggested that the T6SS contributes to niche occupancy by intestinal pathogens such as *S. enterica* serovar Typhimurium, *S. sonnei*, and *V. cholerae*[13–15]. Interbacterial T6SS-mediated competition of these pathogens was mostly tested in vitro using well-characterized laboratory strains as prey (such as *E. coli* MG1655 or DH5α[14]) or mouse-derived bacterial isolates such as *E. coli* (e.g., JB2[13] and WZ1-1 & WZ2-1[15]), *E. cloacae* KL1, *K. oxytoca* TS1, or *K. variicola* KL11 (the latter three strains were species-classified based on 16S rDNA sequencing[13]). In addition, these studies tested the impact of the T6SS-positive pathogens in infant or antibiotic-pretreated mice that had been pre-colonized with these strains. While highly informative, this previous work did not study human commensal isolates. We therefore wondered how *V. cholerae* would interact with members of the human microbiota and considered two hypotheses: (1) *V. cholerae* is able to compete with human gut commensals, given that it can infect human beings; and (2) human gut commensals at least partially protect against T6SS-mediated niche clearance by *V. cholerae*, which would be in line with the high infectious dose that was determined in healthy human volunteer studies[26]. We show that there is a large range in the efficiency of the contact-dependent killing of commensals, whereby certain members of the microbiota are protected from T6SS attacks in an immunity protein-independent manner. This protection occurred by a superior T6SS-mediated killing exerted by members of the *Enterobacter cloacae* complex and by molecular armors made of membrane-tethered capsular polysaccharides of diverse *Klebsiella* isolates. This study therefore contributes to a better understanding of the different mechanisms that underly T6SS-associated interbacterial competition and, accordingly, the maintenance of balanced bacterial communities.

## Results and discussion

**A subset of human gut commensals is protected from *V. cholerae*'s T6SS intoxication**. To start addressing the opposing hypotheses related to *V. cholerae*'s ability/inability to compete with human gut commensal as stated above and to determine how different members of the human gut microbiota might react to *V. cholerae*'s T6SS assaults in vitro, we took advantage of bacterial samples from the Human Gastrointestinal Bacteria Culture Collection (HBC), which is composed of commensal bacteria that were isolated from the gut of healthy human volunteers[27]. We focused our attention on the Gram-negative Enterobacteriaceae (Supplementary Data file 1), as these bacteria are highly abundant in the small intestine[28] where the primary colonization by pathogenic *V. cholerae* occurs[29].

To test *V. cholerae*'s competitiveness against members of the human microbiota, we first needed to choose a strain that had an

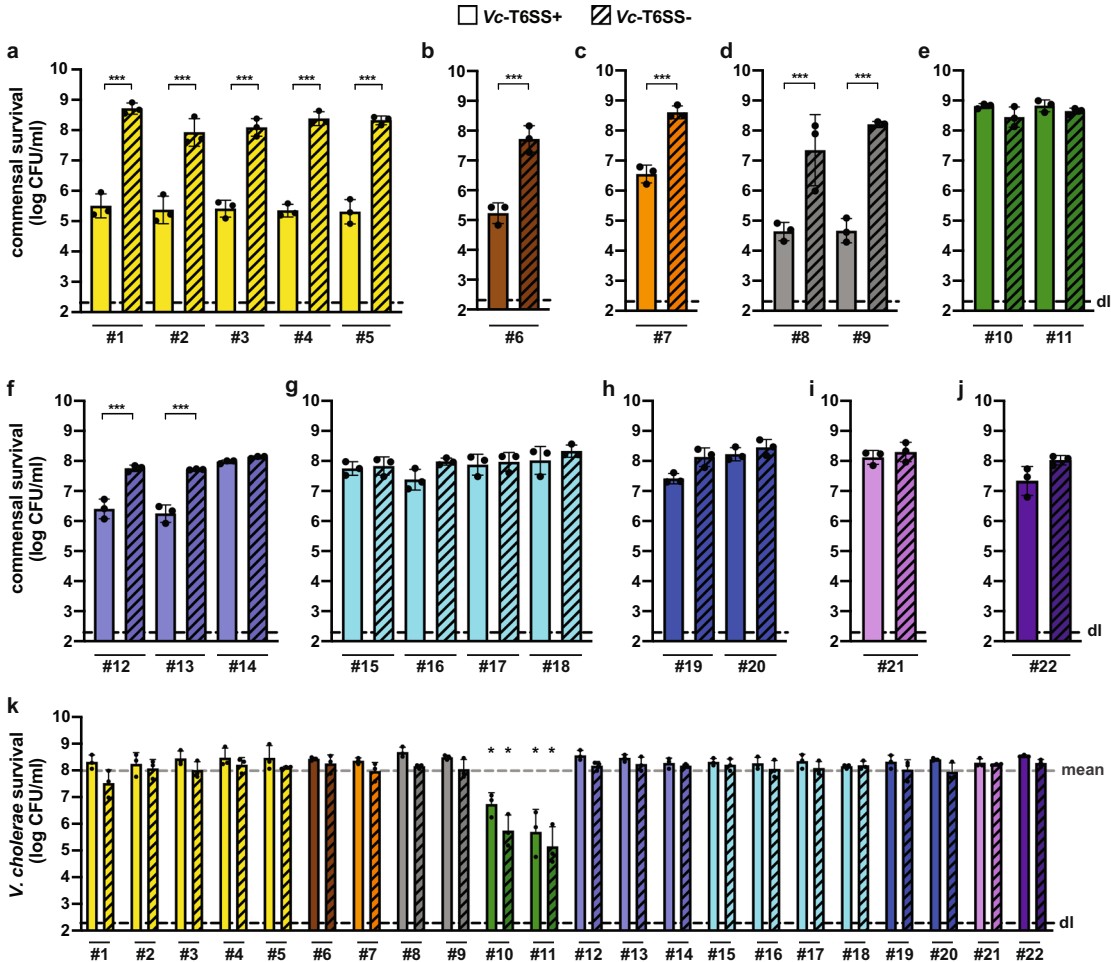

**Fig. 1 A subset of human gut commensals is resistant to *V. cholerae*'s T6SS attacks. a–j** Commensal Enterobacteriaceae show diverse sensitivity to T6SS assaults. Human commensals were tested for survival against toxigenic T6SS+ (WT; plain bars) or T6SS- (ΔvipA; stripped bars) *V. cholerae*. Isolates are grouped by taxa: **a** *Escherichia coli*; **b** *Hafnia alvei*; **c** *Citrobacter freundii*; **d** *Kluyvera cryocrescens*; **e** *Enterobacter cloacae* complex; **f** *Klebsiella michiganensis*; **g** *Klebsiella oxytoca*; **h** *Klebsiella pneumoniae*; **i** *Klebsiella variicola*; and **j** *Klebsiella grimontii*. The commensals' survival is indicated on the Y-axis. Significant differences were determined using a two-sided Student's *t*-test corrected for multiple comparisons. Only significant differences are indicated. ***$p < 0.001$. **k** *V. cholerae* is killed by commensal *Enterobacter* isolates. The survival of T6SS+ (WT; plain bars) or T6SS− (ΔvipA; stripped bars) *V. cholerae* when co-incubated with human commensals was scored. Color code and X-axis labels as in panels (**a–j**). *$p < 0.05$, indicating significant lower survival of *V. cholerae* (T6SS+ & T6SS−) when compared with the mean survival value of all the tested conditions (gray dashed line) as determined by two-sided Student's *t*-tests. Values are derived from three independent experiments and the bars represent the mean (±SD, as defined by the error bars). dl, detection limit, as indicated by the dashed line. Source data underlying all panels are provided in the Source data file.

active T6SS. As members of the pandemic O1 El Tor clade of *V. cholerae* contain a tightly regulated T6SS that is silent under standard laboratory conditions[8,30–32], we used the constitutive T6SS-active toxigenic strain ATCC 25872[32,33] in this study (Supplementary Data file 1). This quorum-sensing-proficient strain is closely related to the O37 serogroup strain V52, which is routinely used in T6SS studies[8,21,23,34]. Notably, both of these O37 serogroup strains carry T6SS and effector/immunity modules identical to those of members of the pandemic O1 El Tor clade.

We first compared the killing ability of the T6SS-active wild-type strain (WT; ATCC 25872) to its T6SS-defective mutant, which lacks the sheath protein VipA-encoding gene, *vipA*. As shown in Fig. 1, the human commensal bacteria attacked by T6SS-positive *V. cholerae* displayed different levels of susceptibility, which correlated strongly with the phylogeny of the strains. Commensal *Escherichia coli*, *Hafnia alvei*, *Citrobacter freundii*, and *Kluyvera cryocrescens* strains were strongly depleted by T6SS-positive *V. cholerae*, while members of the *Enterobacter cloacae* complex (*E. cloacae* and *E. ludwigii*) and the *Klebsiella*

genus (*K. michiganensis*, *K. oxytoca*, *K. pneumoniae*, *K. variicola*, and *K. grimontii*) were either resistant or only slightly impacted by the T6SS assaults under the tested conditions (Fig. 1). Given the large number of resistant strains and that *V. cholerae* cells secrete cocktails of several T6SS effectors into target cells, we concluded that these human commensals must possess immunity protein-independent T6SS-resistance mechanisms. Indeed, the likelihood is very low that all the resistant commensals produce all cognate immunity proteins.

**Strains of the *Enterobacter cloacae* complex kill *V. cholerae* in a T6SS-dependent manner.** As previous work had shown that the colonic microbiota includes T6SS-positive microbes including those of the phylum Bacteroidetes[35], we wondered if the commensals that we had tested for their intoxication by *V. cholerae* (Fig. 1) might also be T6SS-positive. We therefore tested *V. cholerae* survival upon co-incubation with these commensals and observed uniformly high recovery levels, with the exception of

those that had encountered the *Enterobacter* strains (commensals #10 and #11) (Fig. 1k). A previous comparative genomics study revealed two T6SS gene clusters in the *E. cloacae* strain ATCC 13047[36] and this *E. cloacae*-type strain was subsequently shown to constitutively produce its T6SS under in vitro conditions[37]. We therefore assessed whether the commensal *Enterobacter* strains (commensals #10 and #11; Supplementary Data files 1 and 2) also carried T6SS-encoding genes by screening their genomic sequences using the TXSScan program[10]. As a result, we identified a single T6SS cluster (referred to as T6SS-1) for *Enterobacter* strain #10, while commensal #11 possessed two T6SS gene clusters (T6SS-1 and T6SS-2) that differed in their genomic organization (Fig. S1).

To experimentally demonstrate that the *Enterobacter* strains used their T6SS to kill *V. cholerae*, we generated T6SS-inactive mutants of both commensal strains by deleting the gene encoding the essential T6SS core component TssB (ΔtssB). TssB is one of the sheath building blocks and is homologous to VipA in *V. cholerae*. As shown in Fig. 2, WT *Enterobacter* strains #10 and #11 efficiently killed a laboratory strain of *E. coli* (i.e., strain TOP10), while the T6SS-1-impaired mutants (ΔtssB/ΔtssB1) displayed no predatory activity (Fig. 2a) and no Hcp secretion (Fig. 2b). The T6SS-2-impaired mutant (ΔtssB2) of *Enterobacter* strain #11 did not contribute to the antibacterial killing activity under the tested conditions (Fig. 2a). A similar killing pattern was observed when *V. cholerae* served as prey (Fig. 2a), excluding the possibility that the immunity against any putative T6SS-2 activity of commensal #11 was *E. coli*-specific. The lack of interbacterial killing of the T6SS-1 mutants could be complemented by providing *tssB/tssB1* on a plasmid in trans (Fig. S2). Interestingly, the survival of T6SS-positive (Vc-T6SS+) and T6SS-negative (Vc-T6SS−; ΔvipA) *V. cholerae* was affected in a similar manner (Figs. 1 and 2a), suggesting that the *Enterobacter* T6SS killing activity was used in an offensive manner and not as a defensive weapon, as shown for the tit-for-tat strategy of *Pseudomonas aeruginosa*[38]. When we tested the genetically engineered strains of the commensal *Enterobacter*, we observed that *V. cholerae* impaired the survival of the T6SS-1-deficient mutant while the T6SS-2-deficient mutant of commensal #11 was still resistant to intoxication by the *V. cholerae* T6SS (Fig. S2d). Collectively, these data suggest that, at the population level, the commensal *Enterobacter* strains use their T6SS-1 to outcompete the *V. cholerae* population because of their superior killing abilities.

**Presence of T6SS cluster 1 is required but not sufficient for *Enterobacter*'s killing ability.** Since both of the tested *Enterobacter* isolates (commensal #10, an *E. cloacae* species, and commensal #11, an *E. ludwigii* species) showed superior attacking behavior against *V. cholerae* by means of their T6SS-1 cluster, we wanted to verify whether this feature was common among other *E. cloacae* complex strains. This complex is composed of seven species: *E. cloacae*, *E. asburiae*, *E. hormaechei*, *E. kobei*, *E. ludwigii*, *E. mori*, and *E. nimipressuralis*. Of this set, *E. cloacae* and *E. hormaechei* are most frequently isolated from human clinical samples[39]. To investigate the broad T6SS-mediated killing abilities of this complex, we assembled a collection of *E. cloacae* complex strains composed of *E. cloacae*, *E. hormaechei*, and *E. ludwigii* isolates that we obtained from the HBC collection[27], the Baby Biome Study (BBS) collection[40], in which commensal bacteria were isolated from healthy full-term babies, and from the German Collection of Microorganisms and Cell Cultures (DSMZ) (Supplementary Data files 1 and 2). These latter strains were *E. cloacae*-type strain DSM 30054 (equal to ATCC 13047, used in previous studies[36,37]), *E. cloacae* strains DSM 16690 and DSM 26481, and *E. hormaechei* strains DSM 14563, and DSM 30060 (Supplementary Data files 1

and 2). The *E. hormaechei* strains were initially distributed by the DSMZ as *E. cloacae* species, but recently reclassified by the DSMZ (Supplementary Data file 2). As such reclassification of *Enterobacter* species after their whole-genome sequencing seemed to occur frequently, we first verified the identity of the *Enterobacter* commensal isolates based on the assembly of a core-genome-based phylogenetic tree (Supplementary Data file 3) for those isolates for which whole-genome sequencing data were available[27,40] (Supplementary Data file 2). We also (re-)sequenced the *Enterobacter* commensal isolates #10 and #11 and three *Enterobacter* strains that we had obtained from the DSMZ (DSM 30060, DSM 30054, DSM 26481) using a PacBio-based long-read whole-genome sequencing approach (Supplementary Data file 4) and included these genomic data in the analysis.

The phylogenetic tree highlighted five commensal isolates (commensals #8, #9, #12, #13, and #22) that formed a distinctive clade within the tree (Fig. 2c). Based on their 16S rDNA sequence[27], these HBC collection isolates were initially classified as *Enterobacter* species, while their whole-genome sequences reclassified them as *Kluyvera cryocrescens* (commensals #8 and #9), *Klebsiella michiganensis* (commensals #12 and #13), and *Klebsiella grimontii* (commensal #22) species (see "Methods" section). In addition, this core-genome-based phylogeny separated the different *Enterobacter* species into *E. ludwigii*, *E. cloacae*, and *E. hormaechei* (Fig. 2c). Next, we marked the absence or presence of the diverse T6SS cluster(s) (Supplementary Data files 5 to 7) next to the tree, which showed that all of the *Enterobacter* isolates carried a T6SS-1 while the non-*Enterobacter* isolates (#8, #9, #12, #13, and #22) did not (Fig. 2c). The T6SS-2 cluster was detected in all the strains with the exception of the gut commensal isolate #10 as well as the reclassified *Kluyvera* isolates (Fig. 2c). The latter commensals carried a third T6SS cluster (T6SS-3) instead with yet again a different gene order (Fig. S3). The categorization of the identified T6SS clusters into these three different classes (T6SS-1, T6SS-2, and T6SS-3) was further supported by the construction of a phylogenetic tree that was based on the sequences of the conserved T6SS sheaths proteins TssB and TssC encoded by each cluster (Fig. S4).

We next tested the killing capacity of our *E. cloacae* complex collection (including the reclassified *Kluyvera* and *Klebsiella* isolates) against *E. coli* or T6SS-inactivated *V. cholerae* (ΔvipA). These experiments revealed that all *E. cloacae* strains were able to kill both prey species, while the *E. ludwigii* and *E. hormaechei* isolates showed variable competition patterns (Fig. 2d and Fig. S5a, b). Next, we deleted the *tssB1* gene in those predatory strains that were genetically tractable (which, in our hands, was not the case for strain DSM 26481), which abrogated their interbacterial killing capacity (Fig. 2e and Fig. S5c). Interestingly, we observed significant killing of *E. coli* by the WT *E. hormaechei* strain DSM 30060 when compared with its T6SS-1-deficient variant (Fig. 2e), while this was not the case when *V. cholerae* served as prey (Fig. S5c), suggesting a certain degree of target specificity.

The reclassified non-*Enterobacter* isolates displayed no prey killing activity, suggesting that neither the T6SS-2 nor the T6SS-3 could foster interbacterial competition under the tested conditions (Fig. S5a, b). These competition-related differences between the three T6SS systems could reflect variations in the underlying regulatory networks or inactivation of the non-T6SS-1 clusters, as, for instance, suggested by the absence of T6SS-2-mediated Hcp secretion in the *tssB1* mutant of *E. ludwigii* commensal #11 (Fig. 2b). This idea is supported by two recent studies that were published while this work was under review. Briefly, Soria-Bustos and colleagues showed for *E. cloacae* strain ATCC 13047 that the T6SS-2 genes were highly expressed upon growth of the bacteria in Dulbecco's modified Eagle's medium (DMEM) while this was not the case in LB medium (the growth medium of our study). These authors also suggested that the T6SS-2 was implicated in

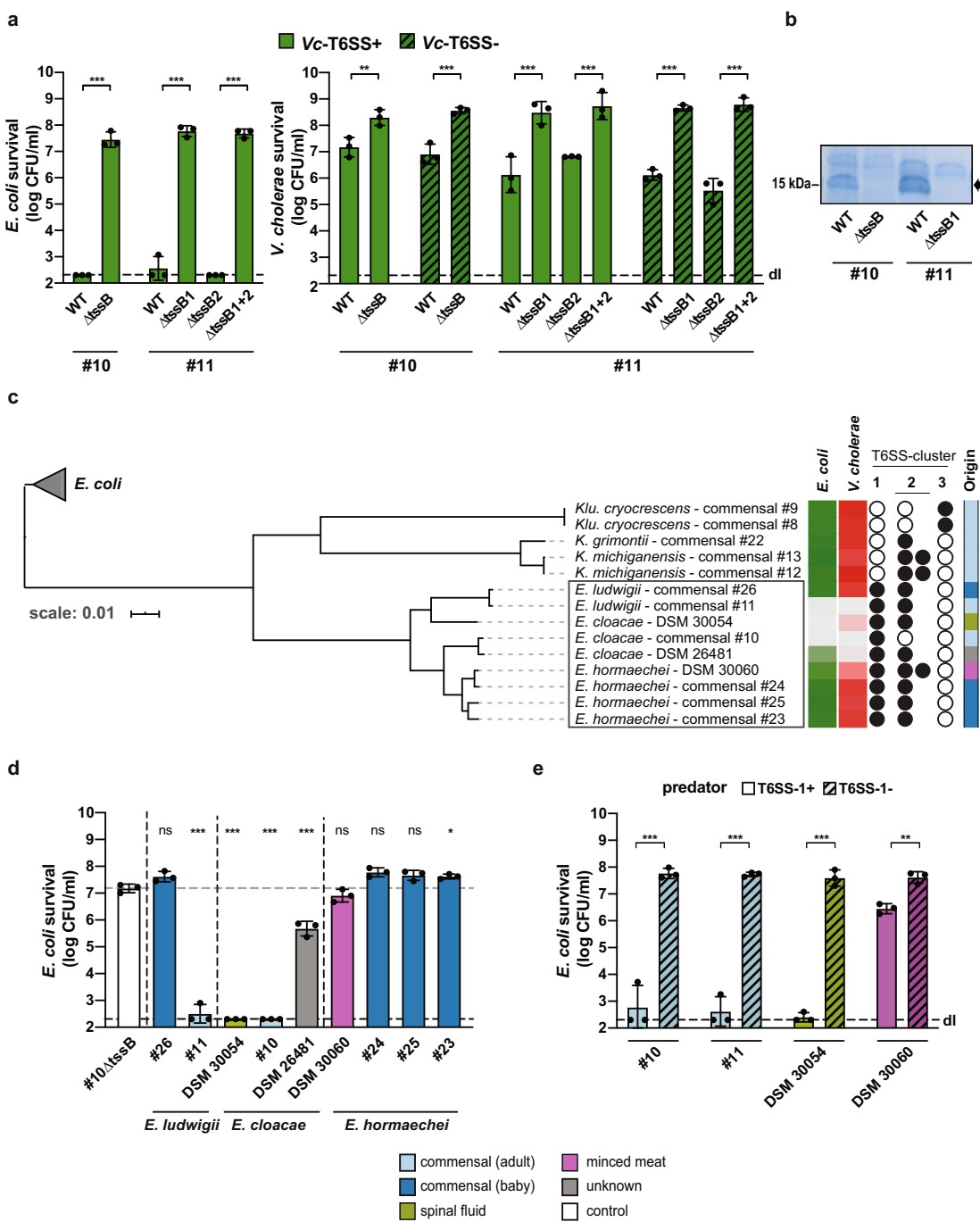

**d** legend:
- commensal (adult)
- commensal (baby)
- spinal fluid
- minced meat
- unknown
- control

biofilm formation and cell adherence and that it contributed to bacterial colonization of the mouse gut in vivo[41]. The finding on the system's functionality should be taken with caution, however, as Donato and colleagues showed that the T6SS-2 of *E. cloacae* strain ATCC 13047 was defective due to a large deletion and the insertion of an IS903 element, which led to the pseudogenization of several T6SS-2 genes (*clpV2, vgrG3, PAAR*, and *tssF2*)[42]. Notably, the sequence of the T6SS-2 cluster in the here-described sequencing data of strain DSM 30054/ATCC 13047 was 100% identical to the one previously reported (accession number CP001918[43]), confirming these pseudogenes.

Interestingly, the T6SS-1 was also detected in most of the non-killing *Enterobacter* strains (Fig. 2c). While the sample number is low, it is interesting to consider the origin of these samples. Indeed, all four T6SS-1-carrying but non-killing *Enterobacter*

isolates were from the microbiota collection of the Baby Biome Study (BBS[40]). It is therefore tempting to speculate that these baby-derived commensals had not yet adapted to the competitive intestinal community. Alternatively, they might depict a target specificity that is beneficial in the maturing phase of the microbiota but may not be functional for in vitro intoxication of *E. coli* or *V. cholerae*. Future studies are therefore required to test these strains' T6SS-1 activity in vitro, their killing capacity against other prey bacteria, and to identify and characterize the strains' effector repertoire.

**The T6SS-1 of *Enterobacter* species is efficient against several human pathogens.** Because several of the T6SS-1-active *Enterobacter* strains were isolated from healthy adults whose microbiota

**Fig. 2 A subset of *Enterobacter* strains kills *E. coli* and *V. cholerae* in a T6SS-1-dependent manner.** Survival of *E. coli* (**a**, **d**, **e**) or *V. cholerae* (**a**) was scored after co-incubation with wild-type (WT) or T6SS-1-/T6SS-2-negative (ΔtssB or ΔtssB1 and/or ΔtssB2) *Enterobacter* commensals #10 and #11 (**a**, **e**) or a collection of *E. cloacae* complex strains (*E. ludwigii*, *E. cloacae*, and *E. hormaechei*) (**d**, **e**), as indicated on the *Y*-axis. Values are derived from three independent experiments and the bars represent the mean (±SD, as shown by the error bars). dl, detection limit, as indicated by the dashed line. Significant differences were determined using a two-sided Student's *t*-test corrected for multiple comparisons (**a**, **e**) and a one-way ANOVA followed by Holm–Sidak's multiple comparison test comparing each strain to the T6SS-deficient control commensal strain (#10ΔtssB; value indicated by the dotted gray line) (**d**). *$p < 0.05$; **$p < 0.01$; ***$p < 0.001$; ns, not significant. **b** Absence of secreted Hcp protein in T6SS-1-negative *Enterobacter* mutants. The commensal *Enterobacter* strains #10 and #11 (WT) and their T6SS-1-deficient mutants (ΔtssB/ΔtssB1), were scored for secreted proteins, which were separated by SDS PAGE and stained using Coomassie blue. The arrow on the right indicates the migration position of the Hcp proteins (~17 kDa), compared to the 15 kDa ladder protein indicated on the left. Representative image (out of three independent experiments). **c** Core-genome-based phylogeny of *E. cloacae* complex strains and reclassified commensal isolates. The tree was rooted with the *E. coli* commensals #1, #2, and #5 as outgroup (gray triangle). The boxed *Enterobacter* strains were tested for interbacterial killing in panel (**d**). Details on the right of the tree (from left to right): First two columns: Summary heatmap of *E. coli* and *V. cholerae* survival, when challenged by the indicated strains as predators. Color scale from light (lowest survival) to dark (highest survival), according to the data provided in supplementary Fig. S5a and S5b. Middle column: Presence (closed circles) or absence (open circles) of the specific T6SS clusters. T6SS clusters were scored as present if at least 10 core T6SS genes were identified in the genome data (detailed information are provided in Supplementary Data file 5). Last column: Origin of isolates according to the legend at the bottom of the figure. Source data underlying all panels are provided in the Source data file.

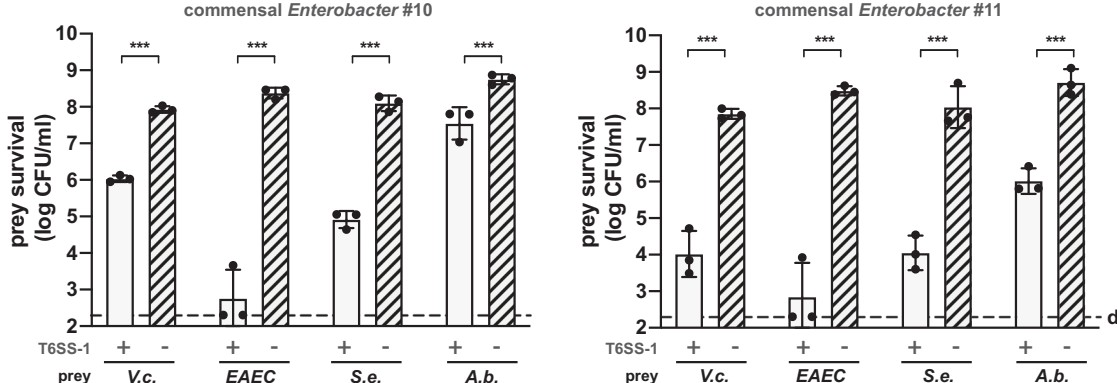

**Fig. 3 *E. cloacae* complex isolates kill pathogenic bacteria in a T6SS-1-dependent manner.** Toxigenic *V. cholerae* (*V.c.*; T6SS-deficient), enteroaggregative *E. coli* (EAEC), *S. enterica* serovar Typhimurium (*S.e.*), or *A. baumannii* (*A.b.*; T6SS-deficient) were co-incubated with commensal *Enterobacter* isolates #10 and #11 and their survival was scored as indicated on the *Y*-axis. Values are derived from three independent experiments and the bars represent the mean (±SD, as defined by the error bars). Significant differences were determined using a two-sided Student's *t*-test corrected for multiple comparisons. ***$p < 0.001$. Underlying source data are provided in the Source data file.

supposedly provide colonization resistance against invading bacteria, we tested whether these commensals were also able to kill other human pathogens. We therefore incubated the *Enterobacter* isolates #10 and #11 with the enteric pathogens *V. cholerae* (strain ATCC 25872), enteroaggregative *E. coli* (EAEC; strain 17-2), *S. enterica* serovar Typhimurium (strain LT2), or *A. baumannii* (strain A118), which is a common colonizer of the gastrointestinal tract[44]. Because these strains also encode T6SSs, we used either T6SS-deficient mutants (for *V.c.* and *A.b.*) or in vitro conditions under which the T6SSs of these strains would not be produced (for EAEC and *S.e.*). As shown in Fig. 3, both commensal *Enterobacter* isolates were able to kill the tested pathogens in a T6SS-1-dependent manner. Notably and in contrast to what we observed for *V. cholerae*, T6SS-positive *A. baumannii* exerted superior killing against the commensal *Enterobacter* strains and remained unaffected by the commensals' T6SS assaults (Fig. S6). Collectively, these data suggest that most *Enterobacter cloacae* and close relatives (e.g., commensal *E. ludwigii* isolate #11) have the capacity to kill selected pathogens using their T6SS-1 machinery. In addition, our work highlights a T6SS-mediated competition hierarchy under the tested conditions. Hence, some commensals and/or pathogens protect themselves from intoxication through the utilization of a superior T6SS, while loss of their T6SS activity makes them vulnerable to assaults from competitors. This finding is in line with a recent study by Perault and colleagues whereby the authors showed that *P. aeruginosa* strains isolated from young

cystic fibrosis patients are protected from *Burkholderia cepacia* complex (Bcc) pathogens in a T6SS-dependent manner[45]. However, during adaption to the host, *P. aeruginosa* strains often acquire T6SS-abrogating mutations, which render them susceptible to T6SS assaults from Bcc strains[45]. The cause of the superiority of the T6SSs in the present study are not known. They could result from the strains' effector repertoire[12], the assembly and/or firing rate of the T6SS machinery, the precise targeting of prey[38,46], the sheath length-dependent force generation, or a combination of all of these features. Apart from promoting their own survival and growth, T6SS-active commensals such as the two commensal *Enterobacter* strains described above, could therefore play a major role in the colonization resistance in human adults toward pathogens.

**Resistance to T6SS assaults correlates with the presence of group I capsules.** While our data unambiguously showed that, at the population level, members of the *Enterobacter cloacae* complex prevent their elimination by *V. cholerae* via T6SS-mediated superior killing, the non-killed *Klebsiella* species did not diminish the *V. cholerae* cell numbers (Fig. 1k). Indeed, most *Klebsiella* species are T6SS-silent under the tested conditions and require specific environmental cues to induce their T6SSs (e.g., change in temperature, oxygen tension, pH, osmolarity[47]). To attempt to explain the survival of *Klebsiella* in the presence of T6SS-positive

*V. cholerae*, we looked to a hallmark of *Klebsiella*—its ability for the production of copious capsular polysaccharide (CPS)[48]. Indeed, CPS is known to confer protection from several external stresses such as phagocytosis, antimicrobial peptides, or components of the human complement[48,49], and we hypothesized that CPS could be playing a role in protecting the tested *Klebsiella* species.

As most Enterobacteriaceae produce capsules belonging to several different capsule groups (such as group I, or Wzx/Wzy-dependent; group IV, or ABC-dependent[50]), we tested whether an association between a specific group and a T6SS-protective phenotype existed. To do so, we first scored the presence of the different capsule groups in all commensal bacteria using CapsuleFinder with the genome sequence of each strain as input[50]. To differentiate between group I capsules versus the secreted polysaccharide colanic acid, which share common biosynthetic pathways, we searched for the *wzi* gene within the identified biosynthetic gene clusters. Wzi is an outer membrane lectin that tethers the capsular polysaccharide to the cell surface[51]. We hypothesized that the capsule could form a physical barrier around the bacteria, and therefore the distinction between surface-tethered versus secreted polysaccharide seemed of prime importance, since common biofilm matrix polysaccharides are often also secreted. Indeed, secreted EPS is a loosely attached structure around the cell body[20], while the membrane-tethered CPS forms a shield around the producing cell[52]. Therefore, for our analysis, *wzi*-carrying strains were considered bona fide group I capsule producers, while commensals that contained the biosynthetic gene cluster but lacked *wzi* were classified as colanic acid producers.

As shown in Fig. 4a, the CapsuleFinder program showed that all commensal strains carried genes encoding for at least one capsule group, while a few isolates, such as the *E. coli* strains (commensals #1–5), *C. freundii* (commensal #7), and the members of the *Enterobacter cloacae* complex (commensals #10–11), carried several capsular–group-encoding biosynthetic clusters. However, none of these strains harbored the *wzi* gene, which strongly suggests that they do not produce a membrane-anchored group I capsule. *Klu. cryocrescens* isolates (commensals #8–9) possessed solely colanic acid production genes, while *H. alvei* (commensal #6) was the only isolate in the tested collection that produced neither colanic acid nor a bona fide group I capsule. Interestingly, all the *Klebsiella* isolates (commensals #12–22) were in silico predicted to synthesize membrane-anchored group I capsules. Finally, none of the commensals carried enzymes to form a group IVe capsule type, which is commonly detected in pathogenic Enterobacteriaceae[50].

To determine whether a specific capsule group was associated with protection from T6SS attacks (Fig. 4a), we searched for associations between the presence of each capsule group and the commensals' T6SS survival phenotype. We excluded *E. cloacae*, since we showed above the causes of its survival. Our analysis showed that survival was positively associated only with the group I capsule ($P < 0.0002$; Wilcoxon test; Supplementary Data file 8), and negatively associated with the colanic acid and ABC capsule. The bacteria with type I capsules do not encode other types of capsules, whereas the other types of capsules are often co-occurring in the same genomes. These results can thus be interpreted as either a positive association between type I capsules and survival that implies accessorily the anti-correlation for the other types, or vice versa. To disentangle this web of associations and pinpoint the most important correlation, we used a stepwise regression analysis to assess if one type of the capsule was sufficient to explain the association of the other types to the T6SS survival phenotype. Indeed, the stepwise regression of the effect on survival of the different capsule groups showed that the only

significant variable was the presence or absence of the group I capsule. The integration of this variable in the regression was sufficient to explain most of the variance in the data ($R^2 = 0.862$, $P < 0.0001$). This supports our hypothesis that this membrane-tethered polysaccharide serves as a protective shield against T6SS-mediated attacks.

**Common *Klebsiella* strains are protected against *V. cholerae*'s T6SS assaults in a capsule-dependent manner.** To test the hypothesis that the group I capsule specifically protects *Klebsiella* species from T6SS intoxication, we took advantage of several well-characterized *Klebsiella* isolates of clinical and environmental origin such as strain 342 (*K. variicola*) as well as strains BJ1-GA, SB617, NJST258-1, and NTUH K2044 (all *K. pneumoniae*; Supplementary Data files 1 and 9). First, we explored genetically engineered strains that were deleted for *wza* using a standard allelic exchange approach[53,54]. Wza is an integral outer membrane lipoprotein and essential for the export of group I capsular polysaccharide[55]. Consequently, *wza⁻* mutants are non-encapsulated[56]. We first confirmed the impaired capsule production by measuring their uronic acid content (Fig. S7a) followed by the imaging of these strains after staining with Indian ink (Fig. S7b). The former method is frequently used as a quantitative measurement for group I capsule production[57]. These experiments confirmed the loss of the group I capsule, so we next assessed the survival of these non-encapsulated mutants in the presence of T6SS attacks exerted by *V. cholerae*. As shown in Figure S7c, the encapsulated strains (CPS+) displayed a significantly higher survival rate compared to the CPS mutants (CPS−), while the survival of *V. cholerae* was indistinguishable after its co-incubation with either the encapsulated or the non-encapsulated *Klebsiella* strains (Fig. S7b). The use of a multivariate linear model accounting for the presence or absence of the capsule and the identity of the strains showed no effect of the capsule on *Klebsiella* survival in the absence of T6SS attacks ($P = 0.56$; *t*-test) and a strong positive effect in the presence of T6SS attacks ($P < 0.0002$). These data indicate a protective role of the *Klebsiella* group I capsule against T6SS-mediated killing by *V. cholerae*. An exception to the powerful capsule-mediated protection effect was observed for *K. pneumoniae* strain NJST258-1, where the survival of the CPS+ strain was also impaired upon T6SS attack. Nonetheless, the CPS− mutant was significantly more sensitive to T6SS assaults than the CPS+ strain (Fig. S7c), illustrating the protective role of the capsule even for this strain. One could speculate that the thickness or the compaction of the capsular material of this strain is lower, thereby providing less protection.

**Encapsulated gut commensal *Klebsiella* are shielded from *V. cholerae*'s T6SS attacks.** As we observed that a capsule-dependent T6SS protection occurred for these well-studied *Klebsiella* isolates, we aimed at testing whether this finding also applied to the *Klebsiella* commensals from the human gut. Hence, we scored their capsule production through uronic acid quantification (Fig. 4b), which supported the in silico predictions. To show causality between capsule production, uronic acid content, and, ultimately, T6SS shielding, we genetically engineered representative strain(s) from each of the commensal *Klebsiella* species by deleting the *wza* gene(s) from their genomes. While members belonging to the *K. pneumoniae* complex (e.g., *K. pneumoniae* and *K. variicola*) carried a single *wza* gene, the commensal *K. oxytoca* and *K. michiganensis* isolates each contained two *wza* copies, with the second copy potentially associated with the production of external structures unrelated to the group I capsule[58]. To determine if the protective activity was

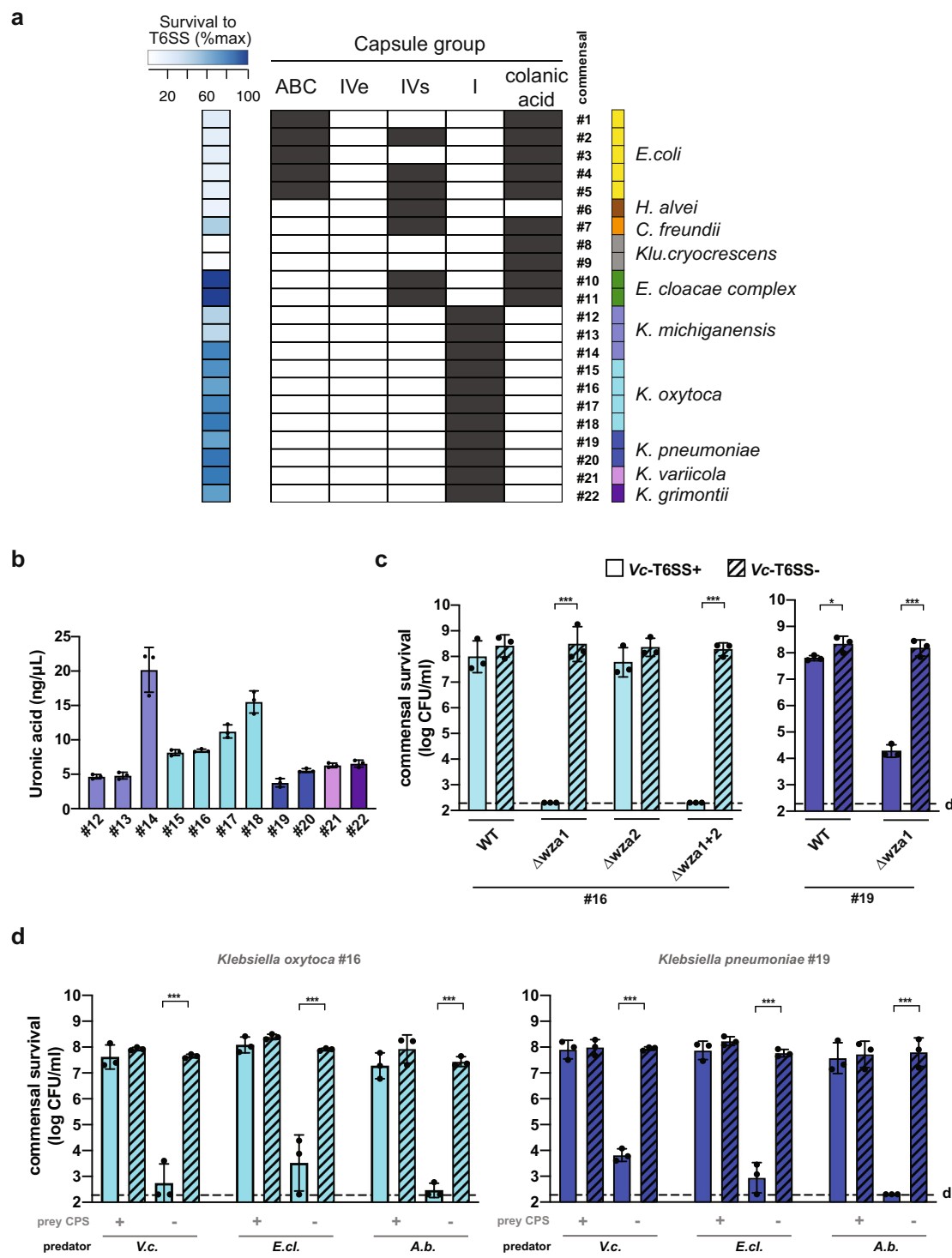

uniquely associated with the group I capsule-specific *wza* gene product, we generated single and double deletion strains of the respective *wza* copies. The uronic acid content decreased (Fig. S8a) in those *wza* mutants that lacked the *wza* gene located within the capsular gene cluster (Fig. S8b) and microscopy of the bacteria stained using Indian ink revealed that these mutants lost their cell-surrounding CPS material (Fig. S8c). Importantly, these *wza* mutants (CPS−) showed a strong sensitivity to T6SS attacks by *V. cholerae*, while their CPS+ parental strains (WT) mostly resisted against the T6SS assaults (Fig. 4c and Fig. S9a), as did

representative *wza*-complemented strains (Fig. S9b) in which CPS was restored (Fig. S9c). Collectively, these data suggest that group I capsules confer a strong protection from *V. cholerae*'s T6SS attacks to commensal *Klebsiella* strains.

**A horizontally acquired *Klebsiella* capsular polysaccharide gene cluster conveys T6SS protection to *E. coli*.** Since impairment of group I capsule production sensitized *Klebsiella* strains to intoxication by *V. cholerae*'s T6SS, we wondered if naturally non-

**Fig. 4 Group I capsule protect commensal *Klebsiella* isolates against *V. cholerae*'s T6SS attacks. a** In silico identification of capsular genes in the commensal collection. Enterobacteriaceae-specific capsule groups (ABC, IVe, IVs, I, colanic acid synthesis) are shown. The blue heatmap (on the left) shows the relative survival values of the T6SS-attacked commensals with 0 and 100 being defined as the lowest and highest log-transformed CFU/ml numbers, respectively (according to the data provided in Fig. 1a–j). Commensal strain numbers and the color code (on the right) are as defined in Fig. 1. **b** The production of group I capsules by the *Klebsiella* gut commensal isolates was assessed by quantification of the strains' uronic acid content. **c, d** Deletion of capsule biosynthesis genes renders commensal *Klebsiella* sensitive to T6SS-mediated intoxication. Representative WT and *wza*-negative *Klebsiella* mutants (commensal *K. oxytoca* #16 and *K. pneumoniae* #19) were tested for survival in the presence of T6SS+ (WT; plain bars) or T6SS– (ΔvipA; stripped bars) *V. cholerae* (panel **c**) or against diverse T6SS-positive (plain bar) and T6SS-negative (stripped bar) pathogens (panel **d**; *V. cholerae* [*V.c.*], *E. cloacae* [*E.cl.*; commensal #10], or *A. baumannii* [*A.b.*; strain A118]). Values are derived from three independent experiments and the bars represent the mean (±SD, as defined by the error bars). dl, detection limit, as indicated by the dashed line. For panels **c** and **d**, significant differences between samples containing T6SS+ and T6SS– predators were determined using a two-sided Student's *t*-test corrected for multiple comparisons. Only significant differences are indicated. *$p < 0.05$; ***$p < 0.001$. Source data underlying all panels are provided in the Source data file.

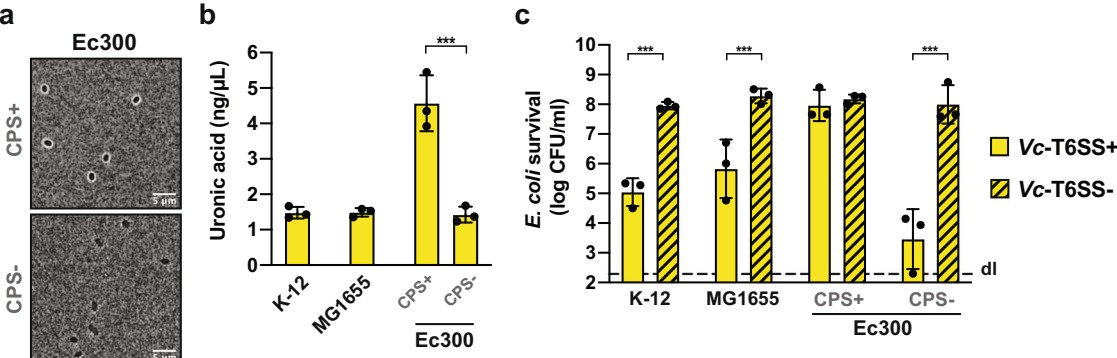

**Fig. 5 Group I capsule-mediated T6SS shielding is a broad protection mechanism that can be horizontally-transferred. a–c** Horizontally acquired capsule biogenesis gene cluster confers protections to *E. coli* against T6SS attacks. **a, b** Capsule visualization and uronic acid quantification of the dog commensal *E. coli* strain Ec300. **a** CPS+ (Ec300; WT) and CPS– (Ec300ΔrfaH) *E. coli* bacteria were imaged after India Ink staining. Representative images are shown. Scale bar, 5 µm. **b** The production of uronic acid was quantified in Ec300, its CPS-minus mutant (ΔrfaH), and two laboratory reference *E. coli* strains as controls. **c** Encapsulated *E. coli* is protected from T6SS assaults by *V. cholerae*. WT (CPS+) and the capsule-minus (CPS–) Ec300 bacteria as well as two reference *E. coli* strains were cocultured with T6SS+ (WT; plain bars) or T6SS– (ΔvipA; stripped bars) *V. cholerae*. Their survival is indicated on the *Y*-axis. **b, c** Values are derived from three independent experiments and the bars represent the mean (±SD, as defined by the error bars). dl, detection limit, as indicated by the dashed line. Significant differences were determined using a two-sided Student's *t*-test corrected for multiple comparisons. Only significant differences are indicated. ***$p < 0.001$. Source data underlying all panels are provided in the Source data file.

encapsulated bacteria could be protected from T6SS attacks if the capsular material was genetically transferred to these bacteria. Fortunately, such a strain already exists in nature in the dog commensal *E. coli* isolate, strain Ec300[59]. It contains an extended *galF-his* region of about 40 kb (compared to ~16 kb in common K-12 *E. coli* strains), which carries a group I capsule biosynthetic gene cluster[49] including *wzi*. It was hypothesized that this capsule-determinant gene cluster was horizontally acquired from *Klebsiella* strain 342 due to the very high sequence similarity between the loci[49]. We investigated this hybrid *E. coli* strain as described above for *Klebsiella* strain 342 (Fig. S7) by visualizing its capsule by microscopy using Indian ink staining (Fig. 5a) and quantifying the strain's uronic acid content (Fig. 5b). The latter was significantly higher than the level of common laboratory *E. coli* strains (strain K-12 and its F-/λ-minus and *rph-1* variant MG1655) or a CPS-negative mutant of Ec300 (ΔrfaH[49]) (Fig. 5b). Consistent with the encapsulation of Ec300, the WT (CPS+) was resistant to T6SS attacks, while the survival of the non-encapsulated mutant (CPS–) as well as the K-12 and MG1655 strains was significantly reduced by *V. cholerae* (Fig. 5c). Together, these experiments show that the group I capsule alone, independently of other genetic traits specific to the genus *Klebsiella*, can provide an efficient protection against T6SS attacks, even when expressed in a heterologous host. Interestingly, a recent study reported that 7% of *E. coli* isolates (with $n = 1194$) derived from Australian freshwater reservoirs had acquired diverse *Klebsiella* group I capsular gene clusters[60]. Based on the

data we present in this study it is possible that those strains might have a fitness advantage when facing T6SS-positive competitors in aquatic environments.

Importantly, these experiments also highlight the difference between group I capsule-mediated protection against T6SS attacks compared to the Rcs pathway-regulated colonic acid production that has been previously described[23]. Indeed, our experiments showed that common laboratory strains of *E. coli* (e.g., K-12 or MG1655), despite their functional Rcs pathway, are insufficiently protected against T6SS attacks when *V. cholerae* secretes toxic effector cocktails instead of a single toxin (e.g., TseH in reference[23]). The T6SS-protective role of colanic acid described by Hersch and colleagues[23] might therefore rely on a different mechanism than the physical shielding expected from a bona fide capsule. Recently, colanic acid was found to contribute to the maintenance of membrane potential during envelope stress, which may be a potential explanation for this protective phenotype[61]. In fact, Pando and colleagues showed that colanic acid-deficient *S. enterica* mutants were more susceptible to ß-lactam antibiotics, such as ampicillin[61], which is consistent with the increased sensitivity of colanic acid-negative *E. coli* when confronted with the peptidoglycan-degrading T6SS toxin TseH of *V. cholerae*[23].

**Encapsulated *Klebsiella* are shielded from a variety of T6SS-producing bacteria.** It is clear from our data that the group I capsules are protective. However, we wondered whether this

molecular armor was specific to T6SS attacks of *V. cholerae* or whether this protection mechanism could be extended to other T6SS-active Gram-negative bacteria. To test this, we compared three T6SS-producing predators, namely toxigenic *V. cholerae* (strain ATCC 25872), *E. cloacae* (commensal #10; see above), and *A. baumannii* (strain A118; Supplementary Data file 1) for their competitiveness against either CPS+ or CPS− *K. pneumoniae*. As shown in Fig. 4d for two representative commensal *Klebsiella* strains and in Fig. S10 for two additional examples as well as for *K. pneumoniae* strains NTUH-K2044 or BJ1-GA as prey, the encapsulated *Klebsiella* strains (CPS+) survived at comparable levels in the presence of the T6SS-positive and T6SS-deficient competitors, while the non-encapsulated (CPS−) derivatives showed a highly significant survival defect when they encountered T6SS-positive attackers. This finding suggests that the barrier exerted by the group I capsular polysaccharide provides a general protective mechanism against T6SS assaults. Of note, while we favor and therefore propose a capsule-mediated physical barrier as protection mechanism, we cannot formally exclude a protective effect of the negative charge exerted by the acidic capsular polysaccharides, which is another commonality among all tested *Klebsiella* isolates and, in general, among group I capsular material[58].

Collectively, in this study, we undertook the first study of *V. cholerae*'s T6SS-dependent interaction with selected human commensal gut Proteobacteria from healthy volunteers. Through our work, we were able to characterize two resistance mechanisms that human commensal gut microbes might use to defend themselves against enteric pathogens. First, we identified a conserved resistance mechanism, whereby commensal *Enterobacter* derived from adult guts carried their own specific and highly potent T6SS (T6SS-1) that prevented their elimination by *V. cholerae*'s T6SS assaults. Second, we identified a so-far undescribed role of the group I capsular polysaccharide of *Klebsiella* species that includes shielding the bacteria from T6SS intoxication. These immunity protein-independent T6SS-resistance mechanisms protect against a broad range of predators and therefore constitute a well-suited defense system. Taken together, our work provides insight into potential bacterial community behavior within the intestinal microbiota and sheds light on how T6SS defense systems might foster colonization resistance against invading pathogens. However, given the in vitro nature of our work, future studies are still needed to address such interactions in vivo, for instance in a humanized mouse model.

Importantly, the identified T6SS protection mechanisms may well extend beyond the gut to other colonization sites in which T6SS-mediated competition plays an important role, such as, for instance, the respiratory tract of cystic fibrosis patients[45] or environmental reservoirs, as suggested above for the encapsulated *E. coli* isolates. A caveat of our study is, however, that the tested commensal isolates were derived from healthy humans, consistent with the state-of-the-art of past volunteer studies[26] and the commonly used animal models[2]. Hence, these isolates might not well reflect the unique microbiome of predisposed people in cholera endemic areas where people are often malnourished and suffering from stunted childhood growth, from co-infections, or from environmental enteric dysfunction[62,63]. Unfortunately, microbiota data on such predisposed populations are still scarce and often lack healthy control groups due to ethical concerns[64]. Nonetheless, our work could serve as a starting point to rationally design T6SS-shielded probiotic strains that are able to restore defective colonization barriers or enhance the barriers' efficiency.

## Methods

**Bacterial strains, plasmids, and growth conditions**. The bacterial strains and plasmids used in this study are listed in Supplementary Data file 1. Unless otherwise stated, the strains were grown aerobically in Lysogeny broth medium (LB;

10 g/l of tryptone, 5 g/l of yeast extract, 10 g/l of sodium chloride; Carl Roth) or on LB agar plates at 37 °C. The following supplements were added at the given concentrations if required: diaminopimelic acid (DAP; 0.3 mM), kanamycin (75 µg/ml), ampicillin (100 µg/ml), carbenicillin (100 µg/ml), streptomycin (100 µg/ml), and chloramphenicol (25 µg/ml). DAP was added as an essential growth supplement for *E. coli* strain MFDpir[65]. S17-1λpir[66] and MFDpir[65] were used for cloning purposes and/or served as donor in bacterial mating experiments. DAP-deficient medium was used to counter-select strain MFDpir after bacterial mating.

**Genetic engineering of strains and plasmids**. DNA manipulations were performed according to standard molecular biology-based protocols[67]. Primers used in this study are listed in Supplementary Data file 10. Enzymes were purchased from the indicated companies and were used as recommended by the manufacturer: Pwo polymerase (Roche), Q5 High fidelity polymerase (New England Biolabs), Expand High Fidelity polymerase (Roche), GoTaq polymerase (Promega), restriction enzymes (New England Biolabs), and T4 DNA ligase (New England Biolabs). Following initial screening by PCR (using bacterial cells as templates), genetically engineered strains and plasmids were verified by Sanger sequencing for their modified regions.

For Enterobacteriaceae, mutants were constructed by in-frame deletion of the target genes using standard allelic exchange approaches. Briefly, upstream and downstream sequences of the respective gene (>500 bp) were PCR amplified using oligonucleotides with 5′-encoded restriction sites and cloned into likewise digested suicide plasmids (pGP704-Sac28[68] or pGP704-Sac-Kan[69]). Ligation was performed overnight at 16 °C with T4 DNA ligase (New England Biolabs). Competent *E. coli* S17-1λpir cells were transformed with the ligated products and transformants were screened for correct insertions using PCR. Positively scored plasmids were verified by Sanger sequencing and then transferred into *E. coli* strain MFDpir[65]. *E. coli* MFDpir served as the donor for conjugation with the respective receptor strains. Bacterial mating was performed for at least 8 h at 37 °C. Single crossover transconjugants were selected on antibiotic-containing agar plates. Next, transconjugants were grown for 16 h at 37 °C and strains with excised plasmids were selected at room temperature on NaCl-free LB agar plates supplemented with 10% sucrose. To confirm the loss of the plasmid, colonies were tested for their antibiotic sensitivity. Deletion mutants were verified by PCR and Sanger sequencing.

**Interbacterial killing assay**. The interbacterial killing assay was performed following a previously established protocol with minor modifications[30]. Briefly, the defined prey and predator cells were harvested after overnight growth, washed, and concentrated to an optical density at 600 nm (OD$_{600}$) of 10 in PBS. Predator and prey were mixed at a ratio of 1:1 and spotted onto membrane filters on pre-warmed LB agar plates. After 4 h of incubation at 37 °C, bacteria were resuspended and serial dilutions were spotted onto selective media that were matched with the characteristics of the bacterial strains (e.g., different antibiotic resistance profiles or growth/no growth on selective media). For each killing assay, the recovery rate of both strains was scored. The Enterobacteriaceae mentioned in Supplementary Data files 1 and 2 were selected on the following selective plates if required: MacConkey agar plates (GMH081; Lucerna Chem AG, Switzerland), Simmons Citrate Agar plates (85463; Sigma-Aldrich) supplemented with 1% inositol (I5125; Sigma-Aldrich), and chloramphenicol-supplemented LB agar plates. *Acinetobacter baumannii* was selected on CHROMagar *Acinetobacter* medium (ACE092(B), CHROMagar, France). The different *V. cholerae* strains were counter-selected and enumerated after spotting on streptomycin-supplemented LB agar plates or Thiosulfate Citrate Bile Salts Sucrose (TCBS) agar plates (Sigma). Recovered colonies were counted to determine the colony-forming units (CFU) per ml. Each experiment was performed three independent times and mean values are shown in the bar graphs. Statistically significant differences were determined on log-transformed data[70]. If no prey bacteria were recovered, the value was set to the detection limit to allow the calculation of the mean and the statistical analyses.

**Hcp secretion assay**. Bacteria were grown overnight under aerobic conditions in 25 ml LB medium at 37 °C. After 10 ml of the culture was harvested by centrifugation, 4.5 ml of the supernatants were sterile-filtered (0.2 µm filter; VWR) and the proteins precipitated using trichloroacetic acid (TCA). The precipitated proteins were washed with acetone before being resuspended in 50 µl of 2X Laemmli buffer. The samples were heated at 95 °C for 15 min before analysis. To visualize the secreted Hcp, the proteins were separated by sodium dodecyl sulfate (SDS)-polyacrylamide gel electrophoresis using precast Mini-PROTEAN TGX Stain-Free gels 8–16% (Bio-Rad). The gels were subsequently stained using InstantBlue™ Coomassie Protein Stain (Expedeon) according to the instructions provided by the manufacturer.

For complementation experiments, bacteria were grown aerobically at 37 °C in LB medium supplemented with kanamycin 75 µg/ml until they reached an optical density at 600 nm of ~0.5. At that point, the cultures were induced by the addition of 0.2% arabinose for 3 h before they were processed as described above.

**Capsule extraction and uronic acid quantification**. The bacterial capsular material was extracted as previously described[71] and the uronic acid content

quantified using the method reported by Blumenkrantz and colleagues[57]. Briefly, an overnight culture was diluted/concentrated to an $OD_{600}$ value of 4. Five hundred microliters of this suspension was mixed with 100 μl of 1% Zwittergent 3–14 detergent (693017–5GM; Sigma; dissolved in 100 mM citric acid, pH 2.0) and heated at 56 °C for 20 min. After incubation, the mixture was centrifuged for 5 min at 20,817 × g and 300 μl of the supernatant was transferred to a new tube. Absolute ethanol was added to a final concentration of 80% and the samples were placed on ice for 20 min. After centrifugation, the pellet was washed with 70% ethanol, dried at 96 °C for 5 min before 250 μl of distilled water was added. The pellet was dissolved during 2 h at 56 °C. Polysaccharides in the isolated capsule material were subsequently quantified by measuring the amount of uronic acid. To do so, 1.2 ml of 0.0125 M tetraborate dissolved in concentrated $H_2SO_4$ was added to 200 μl of the respective sample. The mixture was vigorously vortexed, heated at 96 °C for 5 min, and allowed to cool down again before 20 μl of 0.15% 3-hydroxydiphenol (dissolved in 0.5% NaOH) was added. The tubes were vortexed before 1 ml of the sample was transferred to a cuvette for absorbance measurements at 520 nm. The uronic acid concentration of each sample was determined using a standard curve based on known concentrations of glucuronic acid.

**Capsule visualization by Indian Ink staining**. To visualize the capsule material, 5 μl samples from overnight grown bacterial cultures were mixed with 2 μl of India Ink reagent (BD #261194) on a microscope slide and covered with cover-slip. The bacteria were subsequently imaged in brightfield mode using a Plan-Apochromat 100x/1.4 Ph3 oil objective as part of a Zeiss Axio Imager M2 epifluorescence microscope with an attached AxioCam MRm camera, which was controlled by the ZEN BLUE 2.6 software from Zeiss. Images were analyzed and prepared for publication using ImageJ v2.0.0-rc-69/1.52p.

**Genomic DNA preparation**. Genomic DNA was isolated from 10 ml of LB-grown bacterial overnight cultures using a Qiagen genomic DNA buffer set combined with Qiagen 500/G Genomic-tips. The extraction was performed according to the manufacturer's protocol.

**Long-read whole-genome sequencing**. High molecular weight DNA was sheared in a Covaris g-TUBE (Covaris, Woburn, MA, USA) to obtain an average fragment size of 10 kb. After shearing the DNA, size distribution was checked on a Fragment Analyzer (Advanced Analytical Technologies, Ames, IA, USA). Five hundred nanograms of the DNA was used to prepare a SMRTbell library with the PacBio SMRTbell Express Template Prep Kit 2.0 (Pacific Biosciences, Menlo Park, CA, USA) according to the manufacturer's recommendations. No size selection was applied. The pooled bar-coded libraries were sequenced with v3.0/v3.0 chemistry and diffusion loading on a PacBio Sequel instrument (Pacific Biosciences, Menlo Park, CA, USA) at 600 min movie length, pre-extension time of 120 min, using one SMRT cell 1 M v3.

Genome assembly was performed using CANU 2.0 with the option 'pacbio-raw' and a defined expected genome size of 5 Mbp[72]. The circularization of the genomes was achieved using Circlator v.1.5.5 with default parameter settings[73]. Genes were predicted using Prokka v.1.14.6[74] but, during the submission of the sequenced genomes, reassigned through the NCBI Prokaryotic Genome Annotation Pipeline (PGAP, version 4.11). Sequencing details and NCBI accession numbers are summarized in Supplementary Data file 4.

**Determination of the core genomes of selected strains**. The core-genome reconstruction was done for the initially classified *Enterobacter* strains following a previously published approach[75] using the software OPSCAN v.0.1 (https://bioinfo.mnhn.fr/abi/public/opscan/). Briefly, orthologs were identified as bidirectional best hits using an end-gap-free global alignment between a reference proteome from the group of interest and each of the other proteomes. Hits with <80% amino acid sequence similarity or more than 20% difference in protein length were discarded. As most tested genomes were solely draft assemblies, synteny was not used as a comparison criterium. The core-genome was defined as the shared group of orthologs genes that were identified in each of the comparison against one of the strains (commensal strain #9). The core-genome was judged to consist of 2018 protein-encoding gene sequences. The list of gene families included in the core-genome is in Supplementary Data file 3.

**Phylogenetic reconstruction**. The proteins encoded by the core-genome were individually aligned using the multi-sequence alignment program MAFFT version 7.453[76] with the -linsi parameter. Non-informative regions of the alignment were trimmed using trimAl v1.4.rev15[77] with the automated1 algorithm. The resulting alignments were then concatenated for each genome. The phylogenetic tree was inferred using IQ-TREE v.2.0.4[78] and the best model, LG + F + I + G4, was determined using the option TEST. The robustness of the topology was tested with 1000 rapid bootstrap experiments. The phylogenetic tree was visualized with iTol v5.5.1[79]. Three *E. coli* genomes (ERR2221227, ERR2221250, ERR2221398[27]) were used as outgroups to root the phylogenetic tree. Based on the phylogenetic tree, five commensals (#8, #9, #12, #13, #22) clustered outside the other *Enterobacter* strains. Consistent with this observation, we discovered a recent reclassification of these

strains within the European Nucleotide Archive (ENA) repository, which were now no longer considered as *Enterobacter* strains (Supplementary Data file 2).

**Species reclassification of the commensal isolates**. The taxonomy of the commensal isolates was initially based on their 16S rDNA sequence[27,40]. Here, a whole-genome-based taxonomy approach was used. This taxonomy was either derived from the reclassification of several of the commensal isolates that was recently done by the ENA repository (accession numbers PRJEB23845 and PRJEB22252) or a refined classification using the program Kleborate v. 0.4.0b[80], which was performed as part of this study for the *Klebsiella* strains (including the ENA-reclassified commensals #12, #13, and #22; Supplementary Data file 2). The input for Kleborate were fasta files of the draft/assembled genomes.

**In silico detection of type VI secretion system gene cluster**. Draft genomes of the gut commensal isolates[27,40] were inspected for the presence of putative T6SS operons using the module TXSScan[10] from MacSyFinder v.2[50] and then manually curated. TXSScan was used in ordered replicon mode with the default settings for HMMER options. The identified clusters were represented using the R package genoPlotR v. 0.8.9. The T6SS cluster types defined in this study (T6SS-1, T6SS-2, T6SS-3) are based on the gene organization within the clusters. This classification was further validated through construction of a phylogenetic tree based on the conserved T6SS sheath proteins TssB and TssC encoded in each T6SS cluster. The protein sequences corresponding to TssB and TssC were aligned separately using MAFFT[76] (version 7.453; with -auto parameter) and the alignments were concatenated to infer the TssB-TssC phylogenetic tree with IQ-TREE v.2.0.4[78] using the best model LG + G4, determined using the option TEST. The robustness of the topology was tested with 1000 rapid bootstrap experiments. The phylogenetic tree was visualized with iTol v5.5.1[79].

**In silico detection of capsules biosynthesis genes**. Draft genomes (Illumina-based sequence data) of the gut commensal isolates[27] were inspected for the presence of putative capsule biogenesis operons using CapsuleFinder v.1[50]. This program identifies those capsule types that exist in Enterobacteria (e.g., ABC-dependent, group I or Wzx/Wzy-dependent [Wzy_stricte], group IV subgroups corresponding to the model organism *E. coli* [Group IV_e_stricte], or to the model organism *Salmonella enterica* [Group IV_s_stricte]). CapsuleFinder was used in diderm bacteria mode with the default settings for HMMER options. The absence of additional capsule operon(s) scattered across several contigs was verified manually by altering the contigs' order, which did not result in the detection of additional capsule systems. For two of the *Enterobacter* strains (commensals #10 and #11), the analysis was repeated using their PacBio-sequenced genome assemblies as input; no additional capsule operon(s) were detected, supporting the notion that additional capsule operons were not missed in the incomplete Illumina-based assemblies. The results were further refined by separating the group I capsule types into two subcategories, namely bona fide *wzi*-dependent group I capsules versus *wzi*-lacking colanic acid producers (according to Whitfield[58]). The presence of the different capsule types in each isolate was represented using the R packages ggplot2 v.3.0.0, RColorBrewer v.1.1-2, and the function heatmap.2 from the package gplots v.3.0.1.1.

**Statistical analysis**. Statistical analysis was performed using GraphPad Prism 8.4.2 and 9.0.2 (GraphPad Software, Inc., CA, USA) using log-transformed data. Statistical significance was determined using unpaired two-tailed Student's *t*-test, as indicated in the figure legends. In case of multiple comparisons, the statistical significance was corrected using the Holm–Sidak method. The significance level (α) was set to 0.05 in all cases. $P$ values of all statistically significant results are listed in Supplementary Data file 11. In the graphs, the biologically independent replicates are indicated by circles, while the bars depict the mean of all experiments (±standard deviation, SD).

The association between the capsule types and protection from T6SS attacks was performed using a forward stepwise regression using JMP® 13.2.0 (SAS Institute Inc.), where the criterion of arrest was given by the BIC (similar results for AIC). The significance of the regression was evaluated using an F-test. The effect of each capsule type was tested through non-parametric Wilcoxon test (one-way test using the $\chi^2$ approximation).

**Reporting summary**. Further information on research design is available in the Nature Research Reporting Summary linked to this article.

## Data availability
All data generated or analyzed during this study are included in this published article (and its Supplementary Information and Supplementary Data files). Accession number for whole-genome sequencing data are provided in the text and in Supplementary Data files 2 and 4. The PacBio raw read data of the five whole-genome sequenced *Enterobacter* strains generated in this study have been deposited in the NCBI's Sequence Read Archive (SRA) database under the Bioproject accession number PRJNA640151. Details on the

SRA accession numbers, BioSample accession numbers, and individual genome accession numbers of the de-novo-assembled and circularized genomes are provided in Supplementary Data file 4. Source data are provided with this paper.

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

## Acknowledgements

The authors thank Trevor Lawley and his team for provision of the strains from the Human Gastrointestinal Bacteria Culture Collection (HBC) and the Baby Biome Study (BBS) collection, Jean-Marc Ghigo for sharing *E. coli* strain Ec300 and its *rfaH* mutant, Eric Cascales for *S. enterica* and the EAEC strains, and Bärbel Stecher and Simone Herp for initial discussions on T6SS and microbiota. The authors acknowledge the staff of the Lausanne Genomic Technologies Facility at the University of Lausanne for sample processing and sequencing. This work was supported by a Consolidator Grant from the European Research Council (ERC; 724630-CholeraIndex), the Swiss National Science Foundation (NRP 72 grant 407240_167061), and a grant from the Novartis Foundation for medical-biological research (#18C178) to M.B. M.B. is a Howard Hughes Medical Institute (HHMI) International Research Scholar (#55008726). O.R. received funding from an Agence nationale de la recherche (ANR) JCJC grant [ANR 18 CE12 0001 01 ENCAPSULATION] and the work by E.R., O.R., and A.B. was supported by grants from the Laboratoire d'Excellence IBEID [ANR-10-LABX-62-IBEID] and the Fondation pour la Recherche Médicale [Equipe FRM: EQU201903007835].

## Author contributions

M.B. secured funding, conceived the overall project, and oversaw its implementation; N.F., S.I., and M.B. designed the details of the study; N.F., S.I., L.F.L.R., S.S., C.S., N.V. and M.B. performed the wetlab experiments mostly overseen by N.F.; T.S. performed preliminary experiments; M.G.G. assembled the PacBio-based sequenced genomes; S.I., M.G.G. and E.P.C.R. performed the bioinformatic analyses; O.R., A.B. and E.P.C.R. contributed the CPS+ and CPS− *Klebsiella* clinical and environmental isolate strains; O.R. and E.P.C.R. provided advice on *Klebsiella* biology; S.I. and E.P.C.R. performed the statistical analyses; N.F., S.I. and M.B. analyzed the data; N.F., S.I. and M.B. wrote the manuscript with input from O.R. and E.P.C.R.; M.B. revised the manuscript; all authors approved the final version of the manuscript.

## Competing interests

The authors declare no competing interests.
