## [Peer Review File · Nature Communications]

REVIEWER COMMENTS

Reviewer #1 (Remarks to the Author):

In this work, Flaughnatti and colleagues report differential survival of human commensals when co-incubated with T6SS-active *Vibrio cholerae*. The authors use a genetic approach to show that the observed results are due to one of two mechanisms. First, *Enterobacter* species survive incubation with *Vibrio* by employing their T6SS-1. In contrast, *Klebsiella* species resist killing by *V. cholerae* via their capsule. This study provides insightful information regarding bacteria-bacteria interactions in the context of the human gut. Although the concept that capsule protects against T6SS is not novel, here the authors showed that the protective role of capsule in *Klebsiella* spp is more important than what has been already shown in other species.

Major comments

1) The proposed mechanism of T6SS immunity mediated by simply “building a superior machine” is not particularly a novel finding. This is perhaps best exemplified by the Tit-for-tat behavior of *P. aeruginosa*, whereby T6SS activation and killing actually results from T6SS attacks from a competing bacterium.

2) Overall, the sections of the manuscript focused on characterization of T6SS-1 are weak. The authors should consider the following points:

1. It is unclear why the authors focus so much on the conservation of this system. As the authors point out, and not surprisingly, the presence of T6SS-1 is not sufficient for *Enterobacter*'s killing ability. This is because conservation of a T6SS machine is not informative of T6SS function. Instead, the role of the T6SS is dependent on the effectors that are secreted by this system. Thus, an analysis of the effectors would be much more relevant.

2. The results shown in Figure 2d and 2e are not interpretable because the experiments lack the appropriate controls. To show that bacterial killing is in fact due to T6SS-1, as the authors conclude, the authors must generate T6SS-1 mutants of each strain. The current statistical analyses are not appropriate.

3. Figure 3 does not add much to the story. It simply shows that T6SS-1 of clinical isolates #10 and #11 are bactericidal, which they already showed in Figure 2a and 2b. Moreover, because this assay is done in T6SS-inactive prey strains, and in light of their results from Fig. 1, it is difficult to ascertain the translatable value of these results, as the authors proposed in lines 236-249.

3) Regarding the capsule mechanism of T6SS immunity:

1. It is difficult to interpret the results from Fig. 4 without knowing what *Vibrio* survival looks like when incubated with the various *Klebsiella* lab strains. These strains were not included in Fig. 1.
2. Fig. 4-6: Given that the phenotype of T6SS killing in the capsule mutants is so large (~6 logs), it is critical to show that the capsule defects can be complemented and protection against T6SS can be restored.
3. If the focus of this paper is on human gut commensals, why was the capsule protection against diverse T6SS predators experiment (Fig. 6d) done using the lab strain NTUH?

4) The title "Human gut commensals withstand type VI secretion attacks through immunity-independent mechanisms" can be misleading. The only immunity-independent mechanism presented is capsule protection only for *Klebsiella* species. As it is, it may be interpreted that human gut commensals like members of the *Bacteroidetes* genus employ immunity-independent mechanisms. I suggest a to change the title to describe what is actually demonstrated. "Human gut commensal *Klebsiella* species"...

5) The authors put their emphasis on the type I capsule. However, another commonality among the *Klebsiella* capsules tested is the presence of uronic acids. These negatively charged sugars could be responsible for the effect. With the current data it is not possible to distinguish if the type of capsule or the particular composition of *Klebsiella* capsules is responsible for the protection. This could be investigated using other *Klebsiella* strains lacking GlcA in the structure, or using a different species producing type I capsule.

6) The main finding of the paper is that capsule protects *Klebsiella* from T6SS. Some of the initial experiments with the commensal bacterial species are not very relevant (especially when the defense against T6SS by the capsule also happens in hiper virulent strains). These results are diluting the main findings. Authors could shorten the manuscript considerably to be more focused.

Minor comments

The heat maps in Figures 2c and 4a are not interpretable without a key detailing what which values the different colors represent

Reviewer #2 (Remarks to the Author):

This is a well-written manuscript that identifies important mechanisms of interactions among naturally co-occurring bacterial populations of the human gut. The authors show that although *V. cholerae* can use its T6SS to kill some human commensal strains in culture, others remain resistant to killing. The authors offer evidence that support two mechanisms for resisting T6SS elimination: i) a superior T6SS used in a counter-attack, and ii) protective shielding via capsule. In addition to these important findings, the authors present an impressive amount of work detailing the genotypic and phenotypic variation of diverse human isolates, and how this variation might impact competitive outcomes. Although I found the main conclusions to be well-supported, the exact motivation of the study could be clarified with some strategic re-wording. Below are suggestions for the authors to consider.

Major comments:

1. As stated above, the exact motivation of the study was not always clear to me. In the beginning of the manuscript, the authors point out that “the initial intestinal colonization of the bacterium is not well understood” (lines 27-28) and that “previous studies indicated that the T6SS contributes to niche occupancy by intestinal pathogens”, which lead to the question of “how *V. cholerae* would interact with members of the human microbiota.” For this reader, I was expecting the experimental approaches to directly test the role of *Vc* T6SS in combating gut commensals to help initiate colonization. Although the paper started out this way, it then shifts its focus to how gut commensals resist *Vc* T6SS killing. This transition left me wondering where the paper was headed regarding the role of T6SS in community dynamics. Was it going to focus on how *Vc* uses T6SS to invade or how commensals use T6SS to resist a pathogen like *Vc*? It feels like both points can and are made to some extent, but they could be more clearly connected with some rewording.

2. Lines 161-162: The authors state that the survival of T6SS+ *Vc* and T6SS- *Vc* was affected in a similar manner. Are the authors referring specifically to competitions using isolate #11 or both isolates (#10 and #11)? The data in figure 2b show survival values for the T6SS+ *Vc* that is at least one log higher, compared to the T6SS- *Vc* survival values, suggesting that without the T6SS, *Vc* is more susceptible to T6SS killing by isolate #10. By contrast, the difference in *Vc* survival with isolate #11 appears to be similar regardless of *Vc* T6SS status. Could the authors offer clarification on this point? Additionally, there could be other explanations for the observed data aside from the *Enterobacter* strains using T6SS as a defensive vs offensive weapon. It could also be explained by

strain- or species-specific differences in T6SS sheaths per cell or firing rates. The authors may want to consider these (and possibly other) explanations.

3. Lines 168-169: The authors conclude that the *E. cloacae* strains tested “use their T6SS-1 to attack *V. cholerae*, thereby avoiding their own intoxication.” However, this conclusion taken literally is not supported by the data presented. My interpretation of this statement is that the *E. cloacae* strains use their T6SS-1 to kill a *Vc* target cell before it can be intoxicated. Such a conclusion could only be supported by examining T6SS competition at the single cell level. Because the data presented here are evaluating T6SS competition by quantifying target cell numbers at a population level, it would be best to draw conclusions at the population level as well. Furthermore, the authors show that *E. cloacae* strains are susceptible to *Vc* T6SS intoxication, in the absence of their own T6SS. Therefore, I would suggest the authors consider rewording their interpretation of these data to include the possibility that *E. cloacae* cells may not be inherently protected from *Vc* T6SS intoxication because of their own T6SS-1, but rather that during competition between these two populations of cells, the outcome is such that the *E. cloacae* population is able to outcompete *Vc* because of its superior counterattack abilities.

4. Lines 171-224: Starting at this line the authors begin testing more isolates, some of which are human isolates, a few that are from baby guts, but the remaining are unknown or from locations that have little to do with early colonization of *Vc* in the human gut (ex. spinal fluid, cleaning water, minced meat). It is unclear to me how examining these strains helps to achieve the goal of understanding the role of T6SS interactions in the gut. Therefore, I found these data to be a distraction from the primary goal of this work. However, I do appreciate the amount of effort that it takes to perform the genomic and phenotypic analyses presented. The authors could consider removing these data and publishing them as a separate, small paper that highlights the diversity of T6SS killing ability among such closely-related isolates.

5. The authors identify an interesting link between group I capsule and their gut commensals that are resistant to *Vc* T6SS killing and generate a hypothesis that capsule protects from *Vc* T6SS killing. They then begin to test their hypothesis using different *Klebsiella* isolates (pathogens of the blood, respiratory, and urinary tracts). Because *Vc* and these *Klebsiella* pathogens are unlikely to encounter one another, these experiments initially seemed out of place. However, the authors then go on to directly test their hypothesis in the gut commensals. It is unclear to this reviewer as to why the pathogenic *Klebsiella* strains were used first, but the results are interesting. To better integrate these data, the authors could consider combining them and use the well-characterized *Klebsiella* strains as controls, but then highlight that the discovery of capsule defense may extend beyond the gut to other human colonization sites. For example, a recent paper by Perault and colleagues showed T6SS competition among pathogens of the respiratory tract.
(DOI:<https://doi.org/10.1016/j.chom.2020.06.019>)

6. The methods appear to be missing for how cells were prepared and imaged for capsule staining.

Minor comments

1. Line 134: Consider adding the word “intoxicated” before “target cells” for clarification.

2. Line 168: Consider changing “E. cloacae complex strains” to something simpler like “Enterobacter strains” as the more complete definition of the E. cloacae complex does not come up until line 175.

3. Line 253: The authors state their data show that E. cloacae complex strains “avoid T6SS-depenent killing by V. cholerae by means of counter-attack”. As stated above, more careful wording could be considered as the authors have shown these strains can be killed by Vc T6SS, its only at the population level that they have shown Enterobacter strains can use their own T6SS to prevent their elimination.

4. Figure 4a: The scale bar for survival to T6SS (%max) is just a blank white box for me. I assume this should be shaded from white to dark blue?

5. Line 260 – The authors might consider citing ref 23 here again to help support their hypothesis that CPS could protect Klebsiella from T6SS attack.

6. Lines 295-297. I found the sentence starting with “Since the distribution of the presence of...” to be confusing. Consider rewording if possible.

Reviewer #3 (Remarks to the Author):

SUMMARY

Flagnatti et al. present a well-designed and well-executed series of experiments to determine how the gut microbiome responds to pathogenic invaders like *Vibrio cholerae*, cholera’s causative agent.

The authors took advantage of a strain collection representing the gut microbiome of a healthy person. *V. cholerae* employs a virulence mechanism called the Type VI Secretion System (T6SS) for niche competition. This study provides in-vitro data to support a model in which immunity-independent T6SS is achieved by offensive attack or production of group 1 capsule.

MAJOR COMMENTS

The in-vitro data are clear cut. Commensals with an ability to fight back or protect themselves with a capsule are refractory to *V. cholerae*'s T6SS attack. But do 1x1 in-vitro killing assays properly mimic the complex space and communities of the gut? I suggest to colonize sterile infant or adult mice with their cocktail and determine how a change in T6SS+/capsulated commensals influences colonization of *V. cholerae*. This experiment could be done by barcoding each commensal and pathogen. If their hypothesis is correct, they should see a decrease in *V. cholerae* colonization as T6SS+/capsulated commensals increase.

How well does the gut library represent a gut microbiome with obligate anaerobes in the majority? Are the authors looking at the tip of the iceberg in their competition assays? This might be an issue if the commensals tested depend on unculturable commensal bacteria to put up a commensal barrier. At the very least, the authors should discuss this in detail.

How does the use of microbiota from a healthy individual contribute to our understanding of cholera, considering the unique microbiome in (often malnourished) people living in areas endemic for cholera. Please outline your rationale why you study a microbiome from a healthy person.

MINOR COMMENTS

Figure 1: Competitive index instead of survival should be shown (at least for *Klebsiella* and *Enterobacter cloacae*), which the authors show are resistant to *Vibrio cholerae*.

Figure 2a and 2b: $\Delta tssB1$, $\Delta tssB2$, $\Delta tssB1\Delta tssB2$ complementation are missing.

Figure 2d and 2e: T6SS- controls are missing for strains other than #10 and #11, making it difficult to conclude it is T6SS-mediated killing? It seems that one T6SS- control corresponding to #10 is used as a control for the rest of the strains.

Figure 4c: NJST258-1 does not show protection against *V. cholerae*. The authors say that it could be due to the thickness of the capsule. This claim should be validated by microscopy.

Rebuttal letter by Flaugnatti, Isaac *et al.*

First of all, we would like to thank the editor and the reviewers for taking the time to review our manuscript and for the valuable comments on our work. We have now performed additional experiments to address the reviewers' comments, which, ultimately, strengthened our conclusions. Below, we provide a detailed **summary of the major changes**, followed by a **point-by-point response to the individual comments**.

We very much appreciate the re-review of our study and hope that the editor and all reviewers agree that the initial concerns have been properly addressed in this rebuttal letter and the revised manuscript.

Summary of major changes:

The main manuscript now contains these new or revised figure panels:

- Fig. 2-panel a: Revised figure due to new T6SS-deletion mutant (see explanation below; highly comparable results as for initial submission).
- Fig. 2-panel b: New figure showing T6SS activity as measured by Hcp secretion, as requested by the reviewer(s).
- Fig. 2-panel c: Revised figure after new analyses and manual inspection of T6SS gene clusters in the genomic data and exchange of killing values according to revised panel 2d/Fig. S5 (see explanation below; comparable results as for initial submission).
- Fig. 2-panel d: Revised figure due to new T6SS-deletion mutant (see explanation below; highly comparable results as for initial submission). Statistical values were adjusted according to the reviewer's request.
- Fig. 2-panel e: New figure including T6SS-negative DSMZ-derived *Enterobacter* strains, as requested by the reviewer(s).
- Fig. 3: Revised figure due to new T6SS-deletion mutant of commensal *Enterobacter* #10 and #11 (see explanation below; highly comparable results as for initial submission).
- Fig. 4-panels b-d: Heavily revised figure to address reviewers' comments; data on pathogenic *Klebsiella* were moved to supplementary material; instead, commensal *Klebsiella* strain served as prey (new data). The figure now also includes the capsule-dependent T6SS protection data against other pathogens (e.g., T6SS-positive and T6SS-negative *Enterobacter cloacae* and *Acinetobacter baumannii*; new experiments to include new T6SS mutants; replacing former Fig. 6d).

The supplementary material now contains these new or revised figure panels:

- Fig. S1: Revised figure in which we added commonly used gene names in addition to those annotated using the TXSScan program. Moreover, we Sanger sequenced the *tssF* region of commensal #10 and thereby corrected a sequencing artifact.
- Fig. S2: Heavily revised figure with new data to address the reviewers' comments. We now show the complementation of the T6SS deletion strains of the commensal *Enterobacter* strains #10 and #11 (*tssB*-minus) related to their killing ability against *E. coli* (panel a) and *V. cholerae* (panel c) and their ability to resist killing by *V. cholerae* of the T6SS-positive WT but not its mutant (panel d). In addition, we show T6SS activity of the WT and T6SS-minus as well as the complemented commensal *Enterobacter* strains by scoring their ability to secrete the Hcp protein (new panel b), as requested by the reviewer(s).
- Fig. S3: Revised figure in which we added commonly used gene names in addition to those annotated using the TXSScan program.
- Fig. S4: New figure showing the clustering of the three distinctive T6SS clusters that were detected in *Enterobacter* and the reclassified non-*Enterobacter* species (e.g.,

- Klebsiella* and *Kluyvera* species). The tree is based on concatenated TssB and TssC sequences, as commonly used in the field.
- Fig. S5-panels a-b: Revised figure (former S4) due to new T6SS-deletion mutant (see explanation below; highly comparable results as initial figure). The statistical analysis was changed according to the reviewer's request.
- Fig. S5-panel c: New data. Killing of *Enterobacter* strains (commensal and those obtained from the DSMZ) and their T6SS-negative derivatives against *V. cholerae*.
- Fig. S6: New figure showing the competition between commensal *Enterobacter* strains and *Acinetobacter baumannii* with the latter being either T6SS-positive or T6SS-negative, as requested by the reviewer(s). Survival of either *A. baumannii* (panel a) or the commensal *Enterobacter* isolates (panel b) is shown.
- Fig. S7: Revised figure, which now includes the *Klebsiella* capsule-dependent T6SS protection data using the well-studied pathogenic *Klebsiella* strains, as requested by the reviewer(s) [main figure 4c of initial submission]. Survival of the *V. cholerae* strains is now also shown, according to the reviewer's comment.
- Fig. S8: Revised figure that shows the uronic acid measurement of the representative commensal *Klebsiella* isolates and their capsule-negative mutants. The figure also contains the data provided in former Fig. S6 (e.g., the K serotype prediction and the capsule imaging using Indian Ink staining).
- Fig. S9-panel a: Survival data for additional commensal *Klebsiella* isolates (four different *Klebsiella* species), which complement the representative data shown in the main Fig. 4c.
- Fig. S9-panels b+c: Representative complementation data for different *Klebsiella* isolates (examples for three different *Klebsiella* species), as requested by the reviewer(s) [survival data upon *V. cholerae* predation and capsule visualization are shown in panel b and c, respectively].
- Fig. S10: Revised data (due to new T6SS-deletion mutant of *Enterobacter*; see explanation below; highly comparable results as for initial submission) giving additional examples of capsule-mediated protection against non-*V. cholerae* predators (as in main Fig. 4d). This figure also includes data on two of the well-studied *Klebsiella* strains (NTUH-K2044 and BJ1-GA).

The supplementary material now contains these new or revised tables:

Most tables have been renumbered due to the movement of data within the manuscript, as requested by the reviewer(s).

- Table S1: Heavily revised table to include new strain constructs and plasmids and renaming of strains to match figure labels.
- Table S5: New supplementary table that lists the detected T6SS clusters (in the commensal isolates and DSMZ-derived bacteria; corresponding to Fig. 2c).
- Table S6: New supplementary table showing the nucleotide sequences of the T6SS components that are shown in Table S5.
- Table S7: New supplementary table showing the predicted amino acid sequences of the T6SS components that are shown in Table S5.

Important information on the revised study:

Please note that we were initially unable to complement the T6SS-minus ($\Delta tssB$) mutants of the *Enterobacter* commensals (#10 and #11), which was, most likely, caused by a polar effect. As all the surrounding genes likewise belong to the T6SS cluster gene (as shown in Fig. S1), such a polar effect would be unlikely to affect non-T6SS-related genes. These initial mutants therefore still supported the notion that the observed phenotypes were based on the parental strains' T6SS activity.

Nonetheless, to properly address the referees' comments and to complement the mutation, we reconstructed the *tssB/tssB1* mutants of both commensal *Enterobacter* strains #10 and #11 *de novo*.

These strains were complementable, as now shown in supplementary Fig. S2. To keep consistency throughout the manuscript, we therefore repeated all initial experiments in which the T6SS-negative *Enterobacter* commensal(s) served as controls and replaced them with the data containing the new and complementable mutant strain(s) (all data are again based on three independent biological experiments, according to common scientific practice). Please note that the overall conclusion did not change by this replacement, as the results were almost identical.

Moreover, while addressing the reviewer's comment concerning the requirement for T6SS knockout strains of the DSMZ-derived *Enterobacter* strains to show its causality, we identified an oversight in the previous T6SS cluster analysis of strain DSM 30060. Indeed, upon reanalysis and manual inspection of the strain's genome sequence, we identified a *bona fide* T6SS-1 cluster in this strain. Importantly, deletion of this cluster's *tssB* gene impaired the strain's interbacterial killing ability against *E. coli* as shown in the new Fig. 2e. This was an important finding that supports our hypothesis of T6SS-1-dependent killing and further strengthens the revised manuscript. We apologize for the avoidable mistake in the initial analysis. In accordance with this new finding, we now provide detailed supplementary Tables that indicate the T6SS cluster analyses and the exact nucleotide and amino acid sequences of the corresponding genes/proteins (supplementary Tables S5 to S7).

Point-by-point response to reviewers' comments.

Reviewers' comments (blue); authors' reply (black).

Reviewer n`1 (Remarks to the Author):

#1) In this work, Flaugnatti and colleagues report differential survival of human commensals when co-incubated with T6SS-active *Vibrio cholerae*. The authors use a genetic approach to show that the observed results are due to one of two mechanisms. First, *Enterobacter* species survive incubation with *Vibrio* by employing their T6SS-1. In contrast, *Klebsiella* species resist killing by *V. cholerae* via their capsule. This study provides insightful information regarding bacteria-bacteria interactions in the context of the human gut. Although the concept that capsule protects against T6SS is not novel, here the authors showed that the protective role of capsule in *Klebsiella* spp is more important than what has been already shown in other species.

Authors' reply #1: We appreciate the nice summary by this reviewer. Indeed, we agree that the concept of capsule-dependent protection might not be novel but – to our knowledge – this topic has never been directly addressed in previous studies using bacteria with *bona fide* capsules. Instead, previous work addressed the protective effects of biofilm matrix EPS (exopolysaccharide; e.g., *Vibrio* polysaccharide [VPS] of *V. cholerae*; Toska et al 2018) or colanic acid in the case of *E. coli* (Hersch et al., 2020). For the latter case, we show, however, that WT *E. coli* is insufficiently protected from T6SS intoxication in case *V. cholerae* delivers its entire T6SS effector cocktail. For this reason, we truly believe that the current study is very timely, original, and important.

#2) 1) The proposed mechanism of T6SS immunity mediated by simply “building a superior machine” is not particularly a novel finding. This is perhaps best exemplified by the Tit-for-tat behavior of *P. aeruginosa*, whereby T6SS activation and killing actually results from T6SS attacks from a competing bacterium.

Authors' reply #2: We thank the reviewer for their comment. However, we truly believe that the ability of the *Enterobacter* strains to kill in a T6SS-dependent manner is very different from the Tit-for-tat behavior that has been nicely demonstrated for *P. aeruginosa*. Indeed, as the reviewer is certainly aware of, the “Tit-for-tat” strategy identified by Basler *et al.* corresponds to a “defensive” behavior. In this case, *P. aeruginosa* uses its T6SS to respond to membrane perturbations caused by T6SS attacks or other external insults (such as mating pair formation, Mpf; aka the conjugative machinery) from

competing bacteria (see Basler *et al.*, 2013; Ho *et al.*, 2013). In their studies, the authors showed that competing bacteria that lack a T6SS or Mpf structural genes are no longer attacked by the T6SS of *P. aeruginosa*. This is clearly different from our study in which we show that prey bacteria are killed by *Enterobacter*'s T6SS (commensal strains #10 and #11) even if the prey is T6SS-negative, as shown for *E. coli* TOP10 or T6SS-minus *V. cholerae* (Fig. 2). In light of these results, *Enterobacter*'s T6SS should be considered as an “offensive” rather than a “defensive” T6SS, which distinguishes it from the situation in *P. aeruginosa*. This difference is mentioned as such in the text (“*V. cholerae* was affected in a similar manner (Fig. 1 and Fig. 2a), suggesting that the *Enterobacter* T6SS killing activity was used in an offensive manner and not as a defensive weapon, as shown for the tit-for-tat strategy of *Pseudomonas aeruginosa*”).

#3) 2) Overall, the sections of the manuscript focused on characterization of T6SS-1 are weak. The authors should consider the following points:

1. It is unclear why the authors focus so much on the conservation of this system. As the authors point out, and not surprisingly, the presence of T6SS-1 is not sufficient for *Enterobacter*'s killing ability. This is because conservation of a T6SS machine is not informative of T6SS function. Instead, the role of the T6SS is dependent on the effectors that are secreted by this system. Thus, an analysis of the effectors would be much more relevant.

Authors' reply #3: We thank the reviewer for their comment. We agree that the conservation of the T6SS-1 is not the strongest part of the manuscript but we nonetheless consider it of interest to the reader. Indeed, this part on the different T6SSs was primarily meant to generate a hypothesis based on the correlation between the presence of the T6SS-1 and the *E. coli/V. cholerae* killing phenotype, which was subsequently tested using genetic approaches. As this analysis is solely occupying a single panel in Fig. 2, we don't think it is a major issue to keep it in the manuscript.

We also agree that the nature of the effectors or their combination might be what makes the T6SS so efficient (as discussed in the manuscript). Unfortunately, the effectors present in *Enterobacter* are not well characterized and the prediction based on the effectors from other bacteria might not be reliable and neither identify all effectors nor assign them to the proper T6SS in case a strain carries several T6SS clusters. Indeed, one should consider that even for the well-studied *P. aeruginosa* lab strains new effectors are still being identified after more than 15 years of T6SS research on these strains once novel approaches are used (e.g., Nolan *et al.*, 2019; <https://doi.org/10.1101/733030>). Here, we primarily aim to show that *Enterobacter* outcompetes *V. cholerae* under the tested conditions. Notably, these exact same conditions still allow *V. cholerae* to efficiently kill *E. coli* or other commensal bacteria (Fig. 1) and they also foster the killing of T6SS-negative *Enterobacter*, which strongly supports our claim that *Enterobacter* uses its T6SS to defend itself against attacks by other bacteria.

Collectively, we believe that a detailed prediction and characterization of the effectors of diverse *Enterobacter* strains could be interesting. However, such predictions should be coupled with an in-depth characterization, which is clearly outside the scope of the current study.

#4) 2. The results shown in Figure 2d and 2e are not interpretable because the experiments lack the appropriate controls. To show that bacterial killing is in fact due to T6SS-1, as the authors conclude, the authors must generate T6SS-1 mutants of each strain. The current statistical analyses are not appropriate.

Authors' reply #4: We agree with the reviewer's comment that the data suggest T6SS-dependency but do not unambiguously prove it. We nonetheless considered these data - combined with the T6SS-1-dependent killing phenotypes observed for the *Enterobacter* commensals #10 and #11 - sufficient to state this as a hypothesis.

However, to properly address the reviewer's comment and to show direct causality also for these additional *Enterobacter* strains, we now performed these additional experiments, as suggested. First, we aimed at inactivating the T6SS-1 of those strains that showed full or partial interbacterial killing and for which WGS data were available, namely DSM 30054, DSM 26481 and DSM 30060. Unfortunately, and despite numerous attempts using well-established allelic exchange approaches (including several

controls), strain DSM 26481 appeared as genetically intractable in our hands. We managed, however, to genetically modify the two other strains, and showed that they lost their ability to kill *E. coli/V. cholerae* (new Figs. 2e and S5c). Taken together, these results demonstrate that *Enterobacter* strains DSM 30054 and DSM 30060 likewise use their T6SS-1 for interbacterial competition, similar to the situation described for the commensal *Enterobacter* strains #10 and #11.

Concerning the statistical analysis: we concur that the analysis was slightly unusual, as it compared the values to the average of the surviving bacterial strains. For the revised manuscript, we were more than happy to change the statistic, as requested by the reviewer. The change of the statistical analysis had no impact on the overall conclusion.

#5) 3. Figure 3 does not add much to the story. It simply shows that T6SS-1 of clinical isolates #10 and #11 are bactericidal, which they already showed in Figure 2a and 2b.

Authors' reply #5: We respectfully disagree with this comment. We truly consider these results important, as they show that the *Enterobacter's* killing capacity extends towards other pathogens. Indeed, the purpose of this Figure is to demonstrate that commensal *E. cloacae* T6SS-1 killing capacity is not specific against *V. cholerae* but can be considered as a broader mechanism of colonization resistance against pathogens. We therefore kept these data in the manuscript.

#6) Moreover, because this assay is done in T6SS-inactive prey strains, and in light of their results from Fig. 1, it is difficult to ascertain the translatable value of these results, as the authors proposed in lines 236-249.

Authors' reply #6: We'd like to highlight that only the data for *V. cholerae* and *A. baumannii* as prey were based on T6SS-negative mutants (as mentioned in the legend), which was mostly meant to show that killing by the *Enterobacter* commensals is, in principle, possible. Importantly, for *V. cholerae*, T6SS-positive data were also provided in Fig. 2a. To nonetheless address the reviewer's comment, we now added a new supplementary Figure S6 in which we show the very interesting interaction between the commensal *Enterobacter* strains and T6SS-positive & T6SS-negative *A. baumannii* as prey (panel a) or predator (panel b). The data are discussed in the manuscript accordingly.

#7) 3) Regarding the capsule mechanism of T6SS immunity:

1. It is difficult to interpret the results from Fig. 4 without knowing what *Vibrio* survival looks like when incubated with the various *Klebsiella* lab strains. These strains were not included in Fig. 1.

Authors' reply #7: We fully agree with the reviewer's comment and therefore added the *V. cholerae* survival data in supplementary Fig. S7, panel d.

#8) 2. Fig. 4-6: Given that the phenotype of T6SS killing in the capsule mutants is so large (~6 logs), it is critical to show that the capsule defects can be complemented and protection against T6SS can be restored.

Authors' reply #8: We very much appreciate this comment. We initially did not consider such complementation as essential given that we showed the phenotype in so many different strain backgrounds, making unspecific mutations during strain construction highly unlikely as causative of the observed phenotype in all these mutants. However, in line with good genetics practice, we now performed the requested complementation experiments for three plasmid-accepting commensal *Klebsiella* strains: commensals #14, #16, and #19, which belong to different bacterial species, namely *K. michiganensis*, *K. oxytoca*, and *K. pneumoniae*. These data are provided in the supplementary Fig. S9 (new panel b). We also show the restoration of the capsule in the complemented strains through Indian ink staining and capsule imaging (Fig. S9, new panel c).

#9) 3. If the focus of this paper is on human gut commensals, why was the capsule protection against diverse T6SS predators experiment (Fig. 6d) done using the lab strain NTUH?

Authors' reply #9: Excellent comment indeed. We changed this figure in the revised manuscript by adding new experimental data in which representative commensal *Klebsiella* species were tested against these T6SS-positive predators. The new data fully support our initial claim. The data of two representative commensal strains are now included in Fig 4c and for four additional commensal strains in the supplementary material (Fig. S10, panel a). The experiments using the pathogenic *Klebsiella* strains as prey were still retained and are now presented in panel b of Fig. S10, as we consider those data highly relevant for researchers working on these well-studied (pathogenic) *Klebsiella* strains.

#10) 4) The title “Human gut commensals withstand type VI secretion attacks through immunity-independent mechanisms” can be misleading. The only immunity-independent mechanism presented is capsule protection only for *Klebsiella* species. As it is, it may be interpreted that human gut commensals like members of the Bacteroidetes genus employ immunity-independent mechanisms. I suggest a to change the title to describe what is actually demonstrated. “Human gut commensal *Klebsiella* species”...

Authors' reply #10: We appreciate this comment but respectfully disagree, as the *Enterobacter* survival also occurs independently of any immunity genes. To still address the reviewer's comment and to avoid any link to the Bacteroidetes genus, we now rephrased the title as “Human **commensal gut Proteobacteria** withstand type VI secretion attacks through immunity protein-independent mechanisms”.

#11) 5) The authors put their emphasis on the type I capsule. However, another commonality among the *Klebsiella* capsules tested is the presence of uronic acids. These negatively charged sugars could be responsible for the effect. With the current data it is not possible to distinguish if the type of capsule or the particular composition of *Klebsiella* capsules is responsible for the protection. This could be investigated using other *Klebsiella* strains lacking GlcA in the structure, or using a different species producing type I capsule.

Authors' reply #11: We thank the reviewer for this comment. To our knowledge, group I capsules very frequently contain uronic acid, as stated by Whitfield (2006): “Group 1 capsules are acidic polysaccharides, typically containing uronic acids, and tend to be rather similar in structure. Similar (and occasionally identical) capsules are found in *Klebsiella pneumoniae*”. However, we fully agree that our data do not distinguish “if the type of capsule or the particular composition of *Klebsiella* capsules is responsible for the protection”. We now mentioned this caveat in the text. Notably, both options still support the protective role of the capsule, which is what we try to convey in our study. We therefore do not think that testing any very unusual *Klebsiella* strains, which don't contain uronic acids (if they existed somewhere and had been characterized), would bring anything important to the study, no matter the outcome of such experiments. Indeed, we claim that human commensal *Klebsiella* isolates are significantly protected from T6SS assault by means of their capsule and this claim is well supported by our data, as we tested 11 different isolates plus the five commonly studied (pathogenic) *Klebsiella* strains in this work.

#12) 6) The main finding of the paper is that capsule protects *Klebsiella* from T6SS. Some of the initial experiments with the commensal bacterial species are not very relevant (especially when the defense against T6SS by the capsule also happens in hiper virulent strains). These results are diluting the main findings. Authors could shorten the manuscript considerably to be more focused.

Authors' reply #12: We are somewhat confused by this comment, as the main take-home message of the manuscript is that commensal bacteria are protected from T6SS assaults and the reviewer mentioned this him/herself under point #9 “If the focus of this paper is on human gut commensals, why was the capsule protection against diverse T6SS predators experiment...”. We therefore decided to keep both parts in the manuscript, as we respectfully disagree with the comment that they significantly “dilute the main findings”. However, to make this more acceptable to this reviewer as well as the editor, we now moved the data on the well characterized clinical and environmental *Klebsiella* strains to the supplementary material.

#13) The heat maps in Figures 2c and 4a are not interpretable without a key detailing what which values the different colors represent

Authors' reply #13: We thank the reviewer for their comment. For the initial Fig. 4a, the color key was provided above the column in the pdf document. However, following up on this comment, we realized that in some Acrobat versions, the color code disappeared from the respective box, potentially through conversion of the files during the online submission. We truly apologize for this mistake and hope that the color keys are properly visible in the revised manuscript.

For the initial Fig. 2c (still Fig. 2c in revised manuscript), the color code was provided in the legend (“Summary heatmap of *V. cholerae* and *E. coli* survival when challenged by the respective bacteria. Color scale from light (lowest survival) to dark (highest survival)”). To be precise, the color shading in this panel is based on the killing values provided in the Fig. S5. This information was added to the figure legend in the revised manuscript.

Reviewer n°2 (Remarks to the Author):

#14) This is a well-written manuscript that identifies important mechanisms of interactions among naturally co-occurring bacterial populations of the human gut. The authors show that although *V. cholerae* can use its T6SS to kill some human commensal strains in culture, others remain resistant to killing. The authors offer evidence that support two mechanisms for resisting T6SS elimination: i) a superior T6SS used in a counter-attack, and ii) protective shielding via capsule. In addition to these important findings, the authors present an impressive amount of work detailing the genotypic and phenotypic variation of diverse human isolates, and how this variation might impact competitive outcomes. Although I found the main conclusions to be well-supported, the exact motivation of the study could be clarified with some strategic re-wording. Below are suggestions for the authors to consider.

Authors' reply #14: We thank the reviewer for this excellent summary and the kind words. We very much appreciated the suggestions provided below.

#15) Major comments:

1. As stated above, the exact motivation of the study was not always clear to me. In the beginning of the manuscript, the authors point out that “the initial intestinal colonization of the bacterium is not well understood” (lines 27-28) and that “previous studies indicated that the T6SS contributes to niche occupancy by intestinal pathogens”, which lead to the question of “how *V. cholerae* would interact with members of the human microbiota.” For this reader, I was expecting the experimental approaches to directly test the role of Vc T6SS in combating gut commensals to help initiate colonization. Although the paper started out this way, it then shifts its focus to how gut commensals resist Vc T6SS killing. This transition left me wondering where the paper was headed regarding the role of T6SS in community dynamics. Was it going to focus on how Vc uses T6SS to invade or how commensals use T6SS to resist a pathogen like Vc? It feels like both points can and are made to some extent, but they could be more clearly connected with some rewording.

Authors' reply #15: This is a very helpful comment – thank you very much. Indeed, we started by explaining the state-of-the-art and what has been proposed for *V. cholerae* in the past, namely that the T6SS might contribute to niche occupancy. As we had mentioned in the text, previous work was mostly based on infant animal models without a mature microbiota - sometimes pre-inoculated with commensal mouse *E. coli* strains (such as in Zhao *et al.*, 2018, *Science*). In this latter study, the authors first tested mouse commensal isolates for their T6SS sensitivity *in vitro* but they did not comment on any mouse isolates that were, in fact, resistant to T6SS attacks. On the other hand, adult mice and rabbits are refractory to colonization by *V. cholerae* without previous antibiotic treatment. This colonization resistant suggests that counter measures against T6SS assaults and subsequent colonization might be at play in animals with a mature microbiota – including humans. We therefore considered two opposing hypotheses with respect to human commensals, which we now clarified in the revised manuscript. The primary aim was therefore, to assess the ability of T6SS-positive pathogens to kill human commensal bacteria. Our work ultimately suggests that T6SS-mediated competition by *V. cholerae* could be unsuccessful against certain members of the microbiota, due to immunity protein-independent T6SS resistance mechanisms. We consider this as an important finding that will open up future studies on cholera susceptibility of individual person (obviously, outside the scope of this study). We now revised the manuscript to make the transition from the initial hypotheses of using T6SS to invade a mature microbiota towards the finding of the commensals' T6SS resistance more obvious.

#16) 2. Lines 161-162: The authors state that the survival of T6SS+ Vc and T6SS- Vc was affected in a similar manner. Are the authors referring specifically to competitions using isolate #11 or both isolates (#10 and #11)? The data in figure 2b show survival values for the T6SS+ Vc that is at least one log higher, compared to the T6SS- Vc survival values, suggesting that without the T6SS, Vc is more susceptible to T6SS killing by isolate #10. By contrast, the difference in Vc survival with isolate #11 appears to be similar regardless of Vc T6SS status. Could the authors offer clarification on this point? Additionally, there could be other explanations for the observed data aside from the *Enterobacter* strains using T6SS as a defensive vs offensive weapon. It could also be explained by strain- or species-specific differences in T6SS sheaths per cell or firing rates. The authors may want to consider these (and possibly other) explanations.

Authors' reply #16: This is indeed an interesting observation that we had also considered but that we didn't further discussed, as we weren't sure if the difference was biologically relevant or not. It turns out that in the new set of three biologically independent experiments that we performed for the revision (due to the new Δ tssB mutants of commensals #10 and #11, as explained above), this difference was not maintained. For this reason, the reviewer's comment seems no longer relevant.

#17) 3. Lines 168-169: The authors conclude that the *E. cloacae* strains tested “use their T6SS-1 to attack *V. cholerae*, thereby avoiding their own intoxication.” However, this conclusion taken literally is not supported by the data presented. My interpretation of this statement is that the *E. cloacae* strains use their T6SS-1 to kill a Vc target cell before it can be intoxicated. Such a conclusion could only be supported by examining T6SS competition at the single cell level. Because the data presented here are evaluating T6SS competition by quantifying target cell numbers at a population level, it would be best to draw conclusions at the population level as well. Furthermore, the authors show that *E. cloacae* strains are susceptible to Vc T6SS intoxication, in the absence of their own T6SS. Therefore, I would suggest the authors consider rewording their interpretation of these data to include the possibility that *E. cloacae* cells may not be inherently protected from Vc T6SS intoxication because of their own T6SS-1, but rather that during competition between these two populations of cells, the outcome is such that the *E. cloacae* population is able to outcompete Vc because of its superior counterattack abilities.

Authors' reply #17: We fully agree with the reviewer's comment and are thankful for the excellent suggestion, which we adopted exactly as suggested (“Collectively, these data suggest that, at the population level, the commensal *Enterobacter* strains use their T6SS-1 to outcompete the *V. cholerae* population because of their superior killing abilities”).

#18) 4. Lines 171-224: Starting at this line the authors begin testing more isolates, some of which are human isolates, a few that are from baby guts, but the remaining are unknown or from locations that have little to do with early colonization of Vc in the human gut (ex. spinal fluid, cleaning water, minced meat). It is unclear to me how examining these strains helps to achieve the goal of understanding the role of T6SS interactions in the gut.

Therefore, I found these data to be a distraction from the primary goal of this work. However, I do appreciate the amount of effort that it takes to perform the genomic and phenotypic analyses presented. The authors could consider removing these data and publishing them as a separate, small paper that highlights the diversity of T6SS killing ability among such closely-related isolates.

Authors' reply #18: We appreciate the reviewer's comment but do not fully share their opinion. Instead, we believe that extending the finding to other *Enterobacter* strains broadens the scope, as *Enterobacter* strains are also relevant in a gut-non-associated context. We therefore retained this part in the revised manuscript.

#19) 5. The authors identify an interesting link between group I capsule and their gut commensals that are resistant to Vc T6SS killing and generate a hypothesis that capsule protects from Vc T6SS killing. They then begin to test their hypothesis using different *Klebsiella* isolates (pathogens of the blood, respiratory, and urinary tracts). Because Vc and these *Klebsiella* pathogens are unlikely to encounter one another, these experiments initially seemed out of place. However, the authors then go on to directly test their hypothesis in the gut commensals. It is unclear to this reviewer as to why the pathogenic *Klebsiella* strains were used first, but the results are interesting. To better integrate these data, the authors could consider combining them and use the well-characterized *Klebsiella* strains as controls, but then highlight that the discovery of capsule defense may extend beyond the gut to other human colonization sites. For example, a recent paper by Perault and colleagues showed T6SS competition among pathogens of the respiratory tract. (DOI:<https://doi.org/10.1016/j.chom.2020.06.019>)

Authors' reply #19: This is again an interesting and very relevant comment for which we thank the reviewer. Indeed, we could limit our findings to the commensal *Klebsiella* strains. However, given the extensive work on pathogenic *Klebsiella* and their role as ESKAPE pathogens, we think that our data are also highly relevant to people working in these fields. This notion seems to be supported by the reviewer's comment "but the results are interesting".

As for the presentation, we do not see a major advantage of combining the panels addressing the pathogenic versus commensal strains into a single figure, which would come at the risk of overloading the figure. However, to accommodate the reviewer's suggestion, we moved the data on the well-studied *Klebsiella* into the supplementary data and provide data for the commensal strains in the main figure.

We appreciate the comment that the discovery of capsule defense may extend beyond the gut to other human colonization sites, which we indeed discussed insufficiently in the initial manuscript. We now revised the manuscript and mentioned the important study on T6SS competition among pathogens in the CF lung by Peggy Cotter's lab in two parts of the manuscript, as suggested.

#20) 6. The methods appear to be missing for how cells were prepared and imaged for capsule staining.

Authors' reply #20: We truly apologize for this oversight; we now included the method in the revised manuscript.

#21) Minor comments

1. Line 134: Consider adding the word "intoxicated" before "target cells" for clarification.

Authors' reply #21: Done, as suggested.

#22) 2. Line 168: Consider changing “E. cloacae complex strains” to something simpler like “Enterobacter strains” as the more complete definition of the E. cloacae complex does not come up until line 175.

Authors’ reply #22: Great suggestion, which we implemented, as requested.

#23) 3. Line 253: The authors state their data show that E. cloacae complex strains “avoid T6SS-dependent killing by V. cholerae by means of counter-attack”. As stated above, more careful wording could be considered as the authors have shown these strains can be killed by Vc T6SS, its only at the population level that they have shown Enterobacter strains can use their own T6SS to prevent their elimination.

Authors’ reply #23: We revised this part of the manuscript (in line with the authors’ reply #17).

#24) 4. Figure 4a: The scale bar for survival to T6SS (%max) is just a blank white box for me. I assume this should be shaded from white to dark blue?

Authors’ reply #24: Please see authors’ reply #13. We apologize for this file conversion mistake, which has been fixed in the revised manuscript.

#25) 5. Line 260 – The authors might consider citing ref 23 here again to help support their hypothesis that CPS could protect Klebsiella from T6SS attack.

Authors’ reply #25: The mentioned study (ref #23) addresses the biofilm matrix exopolysaccharide of *V. cholerae* (e.g., cell-non-attached polysaccharide). This study was properly cited in the introduction (hence, ref #23). Given that this specific study does not address *bona fide* capsules, we considered it out of place at the indicated position of the text.

#26) 6. Lines 295-297. I found the sentence starting with “Since the distribution of the presence of...” to be confusing. Consider rewording if possible.

Authors’ reply #26: We revised the sentence, as suggested by the reviewer. Thank you for pointing out this confusing sentence.

Reviewer n°3 (Remarks to the Author):

#27) SUMMARY

Flagnatti et al. present a well-designed and well-executed series of experiments to determine how the gut microbiome responds to pathogenic invaders like *Vibrio cholerae*, cholera’s causative agent. The authors took advantage of a strain collection representing the gut microbiome of a healthy person. *V. cholerae* employs a virulence mechanism called the Type VI Secretion System (T6SS) for niche competition. This study provides in-vitro data to support a model in which immunity-independent T6SS is achieved by offensive attack or production of group 1 capsule.

Authors’ reply #27: We thank the reviewer for the excellent summary of our work.

#28) MAJOR COMMENTS

The in-vitro data are clear cut. Commensals with an ability to fight back or protect themselves with a capsule are refractory to *V. cholerae*'s T6SS attack. But do 1x1 in-vitro killing assays properly mimic the complex space and communities of the gut?

I suggest to colonize sterile infant or adult mice with their cocktail and determine how a change in T6SS+/capsulated commensals influences colonization of *V. cholerae*. This experiment could be done by barcoding each commensal and pathogen. If their hypothesis is correct, they should see a decrease in *V. cholerae* colonization as T6SS+/capsulated commensals increase.

Authors' reply #28: We very much appreciate this comment that the *in vitro* data are clear cut; this was our intention. We also fully agree that *in vitro* pairwise competition, as is currently state-of-the-art in T6SS research, is only a starting point to better understand complex bacterial communities. However, we consider the knowledge that commensal bacteria can resist T6SS assaults in immunity protein-independent manners an important first step into this direction.

The suggestion to colonize sterile infant or adult mice is certainly interesting but entirely outside the scope of this study for several reasons (including the lack of bioethical approval for animal experimentations, the requirement for the *de novo* setup of a mouse model, etc). Most importantly, the commensal isolates are derived from humans and might not do well in a heterologous host such as the mouse. Thus, while a successful experiment might provide support in favor of our hypotheses, a failed experiment would certainly not disprove our claims, given that there could be so many maladapted parameters in such an artificial setup (e.g., sterile = germ-free mice, or those treated with antibiotic; human commensals and not mice commensals; composition of the commensal cocktail and ratio between species; MOI of commensals and *V. cholerae*; timing; etc).

#29) How well does the gut library represent a gut microbiome with obligate anaerobes in the majority? Are the authors looking at the tip of the iceberg in their competition assays? This might be an issue if the commensals tested depend on unculturable commensal bacteria to put up a commensal barrier. At the very least, the authors should discuss this in detail.

Authors' reply #29: This is a very interesting question. Indeed, common gut libraries might underrepresent obligate anaerobes for obvious reason (e.g., oxygen sensitivity for growth or survival). With respect to our study: as *V. cholerae* is known to colonize the small intestine, we do not necessarily expect that it would interact much/at all with obligate anaerobes. Instead, as we described in the text, we focused on Gram-negative Enterobacteriaceae, which are known to be present in the small intestine. Importantly, we chose this culture collection, as the authors of the published work provided genome sequencing data for the described commensals (Forster *et al.*, 2019; Shao *et al.*, 2019). Given that we primarily focused on those species that were resistant, it seems that they don't require additional commensal-derived signals to put up this barrier. For those species that are killed by *V. cholerae* (*E. coli*, *Hafnia*, etc), we can indeed not exclude that they might behave differently *in vivo*. For this reason, we now revised the text and added "under the tested conditions". We also mentioned in the abstract that we only tested a selection of human commensal isolates and that we did so *in vitro*.

In a way our findings might indeed only be the tip of the iceberg and we expect a plethora of future studies showing similar defense strategies in diverse bacterial species and communities.

#30) How does the use of microbiota from a healthy individual contribute to our understanding of cholera, considering the unique microbiome in (often malnourished) people living in areas endemic for cholera. Please outline your rationale why you study a microbiome from a healthy person.

Authors' reply #30: This is again a very important point, which we very much appreciate. Notably, past volunteer studies and also animal models likewise do not reflect people in cholera endemic areas. Yet, healthy human volunteers can be infected – even though this requires an incredibly high infectious dose. These studies are commonly cited in the field, as they are in our manuscript. The underlying question of our study, which we developed in the manuscript, addressed the observations that on one

hand *V. cholerae* is able to compete with human gut commensals, given that it can infect human adults, while, on the other hand, healthy people are highly refractory to infection. The rationale for the latter was therefore that the microbiota might play a role in defense in case *V. cholerae* would use its T6SS, as has been suggested (Zhao et al., 2018).

More generally, studies on malnourishment and/or co-infections (for instance, with helminth) are indeed very rare or even non-existing in the cholera pathogenesis field. This is certainly something that should be addressed in the future. There is, however, some recent work on the microbiota of people in cholera endemic areas of the world (for instance, Chen *et al.*, 2020 on environmental enteric dysfunction in Bangladeshi children but also Vonaesch *et al.*, 2018 on stunted childhood growth in sub-Saharan African children). Unfortunately, these data cannot be used to support or refuse our claims, as these studies used 16S sequencing data that do not provide any detailed insight into the commensals' genome content (e.g., presence/absence of T6SS cluster(s) in *Enterobacter* strains or capsule cluster(s) in *Klebsiella* species). We nonetheless discuss this topic in the revised manuscript and disclose the caveat of using samples from human volunteers.

#31) MINOR COMMENTS

Figure 1: Competitive index instead of survival should be shown (at least for *Klebsiella* and *Enterobacter cloacae*), which the authors show are resistant to *Vibrio cholerae*.

Authors' reply #31: We thank the reviewer for the comment but respectfully disagree. Indeed, the way we show the data is how it is commonly done in the field and in all our previous publications (see, for instance, Borgeaud *et al.*, 2015, *Science*). Importantly, in Fig. 1 and several other figures, the values for both the prey and the predator are provided (input cell numbers are always adjusted to the same levels).

#32) Figure 2a and 2b: $\Delta tssB1$, $\Delta tssB2$, $\Delta tssB1\Delta tssB2$ complementation are missing.

Authors' reply #32: We thank the reviewer for this comment. We did not consider these experiments as essential in the initial manuscript, as these specific *tssB* genes were situated in the middle of the large T6SS-1 clusters. For this reason, (polar) effects on neighboring genes would only affect other T6SS genes. It is correct, however, that we could not fully exclude that secondary mutation(s) might have occurred during strain construction in all those mutant strains and could therefore cause the killing deficiency. For this reason, we performed the requested complementation experiments and included them in the revised manuscript (Fig. S2).

Please note that, as explained at the beginning of this rebuttal letter, the initial *tssB* knockout mutants were indeed not complementable (most likely due to a polar effect). Consequently, we reconstructed new *tssB/tssB1* knockout strains. The outcome on interbacterial competition was, as expected, almost identical to the data provided for the initial *Enterobacter tssB/tssB1* mutants. For consistency we replaced all experiments that included any of the *tssB* mutant(s) with data using these new mutant(s), as explained in detail at the beginning of this rebuttal letter.

#33) Figure 2d and 2e: T6SS- controls are missing for strains other than #10 and #11, making it difficult to conclude it is T6SS-mediated killing? It seems that one T6SS- control corresponding to #10 is used as a control for the rest of the strains.

Authors' reply #33: We thank the reviewer for this comment. Indeed, in the initial manuscript, we did not delete the T6SS of all other killing-proficient strains, as the focus of our study was primarily on the commensal isolates. We now followed the reviewer's critique and tried to also delete the T6SS genes in the other killing-proficient *Enterobacter* strains for which genome sequences were available (e.g., DSM 30054, DSM 30060 and DSM 26481). Unfortunately, strain DSM 26481 was genetically intractable in our hands, despite several attempts containing many controls. However, we successfully modified strains DSM 30054 and DSM 30060 and the data fully support our claims (now presented in Fig. 2e and S5c).

#34) Figure 4c: NJST258-1 does not show protection against *V. cholerae*. The authors say that it could be due to the thickness of the capsule. This claim should be validated by microscopy.

Authors' reply #34: We thank the reviewer for this comment. Indeed, strain NJST258-1 is not entirely protected against T6SS assaults; however, there is still a ~10-fold protection compared to the unencapsulated mutant, which is statistically significant, as indicated in the figure. For this reason, we do not think that this finding contradicts our claim on CPS-dependent protection.

Unfortunately, the Indian Ink staining method is not meant to quantitatively measure capsule thickness. We therefore tried EM-based methods which were, however, unsuccessful due to Biosafety regulations that require extensive fixation of all samples before they can leave the dedicated BSL 2 laboratories. Collectively, we consider such thickness measurements outside the scope of the current study, as all commensal *Klebsiella* strains showed very striking and highly significant capsule-dependent protection phenotypes.

REVIEWERS' COMMENTS

Reviewer #1 (Remarks to the Author):

The authors have made a considerable effort to address the main issues raised by the three reviewers. As a result, the manuscript has been very much improved. Most of my experimental concerns have been properly addressed. I still disagree with some of the interpretations and with the way the work is presented...but this is not my paper! I believe this important work can be published as is, and the scientific community will take care of the rest. Therefore, I recommend acceptance.

Reviewer #2 (Remarks to the Author):

The authors provided careful and thoughtful responses to the reviewer comments and made many changes to the original manuscript that have strengthened their findings.

Reviewer #3 (Remarks to the Author):

SUMMARY

The authors replied to major and minor comments in detail and made changes in the revised manuscript. The authors also added two suggested key experiments to the revised manuscript: (1) knockout mutant in T6SS required as T6SS- controls. The authors successfully constructed the T6SS-controls in DSM 30054 and DSM 30060 but not in DSM 26481. (2) the authors complemented *tssB/tssB1* knockout strains in trans with an episomal element. Unfortunately, infant mice infection, and EM-microscopy to measure capsule thickness were not considered. Although the authors agree that the suggested experiments would improve this study, they did not pursue them mostly for biosafety reasons. The authors state that the animal model has many limitations. While I agree with this statement, it should also be applied to nutrient agar plates exclusively used in this study. I suggest the author find collaborators who can conduct the animal studies. I think in-vivo studies are crucial (even as supplementary figures) for a publication. Other concerns remain untouched in this version and are listed below in italics.

#28) MAJOR COMMENTS

The in-vitro data are clear cut. Commensals with an ability to fight back or protect themselves with a capsule are refractory to *V. cholerae*'s T6SS attack. But do 1x1 in-vitro killing assays properly mimic the complex space and communities of the gut?

Sterile infant or adult mice need to be infected with their cocktail and determine how a change in T6SS+/capsulated commensals influences colonization of *V. cholerae*. This experiment could be done by barcoding each commensal and pathogen. If the hypothesis is correct, they should see a decrease in *V. cholerae* colonization as T6SS+/capsulated commensals increase.

Authors' reply #28: We very much appreciate this comment that the in-vitro data are clear cut; this was our intention. We also fully agree that in-vitro pairwise competition, as is currently state-of-the-art in T6SS research, is only a starting point to better understand complex bacterial communities. However, we consider the knowledge that commensal bacteria can resist T6SS assaults in immunity protein independent manners an important first step into this direction.

The suggestion to colonize sterile infant or adult mice is certainly interesting but entirely outside the scope of this study for several reasons (including the lack of bioethical approval for animal experimentations, the requirement for the de novo setup of a mouse model, etc). Most importantly, the commensal isolates are derived from humans and might not do well in a heterologous host such as the mouse. Thus, while a successful experiment might provide support in favor of our hypotheses, a failed experiment would certainly not disprove our claims, given that there could be so many maladapted parameters in such an artificial setup (e.g., sterile = germ-free mice, or those treated with antibiotic; human commensals and not mice commensals; composition of the commensal cocktail and ratio between species; MOI of commensals and *V. cholerae*; timing; etc).

Reviewer's comment: Authors agree with the idea that the mice experiments is interesting. However, authors decided not to pursue this line of investigation (see Summary statement).

#29) How well does the gut library represent a gut microbiome with obligate anaerobes in the majority? Are the authors looking at the tip of the iceberg in their competition assays? This might be an issue if the commensals tested depend on unculturable commensal bacteria to put up a commensal barrier. At the very least, the authors should discuss this in detail.

Authors' reply #29: This is a very interesting question. Indeed, common gut libraries might underrepresent obligate anaerobes for obvious reason (e.g., oxygen sensitivity for growth or survival). With respect to our study: as *V. cholerae* is known to colonize the small intestine, we do

not necessarily expect that it would interact much/at all with obligate anaerobes. Instead, as we described in the text, we focused on Gram-negative Enterobacteriaceae, which are known to be present in the small intestine. Importantly, we chose this culture collection, as the authors of the published work provided genome sequencing data for the described commensals (Forster et al., 2019; Shao et al., 2019). Given that we primarily focused on those species that were resistant, it seems that they don't require additional commensal-derived signals to put up this barrier. For those species that are killed by *V. cholerae* (*E. coli*, *Hafnia*, etc), we can indeed not exclude that they might behave differently in vivo. For this reason, we now revised the text and added "under the tested conditions". We also mentioned in the abstract that we only tested a selection of human commensal isolates and that we did so in vitro. In a way our findings might indeed only be the tip of the iceberg and we expect a plethora of future studies showing similar defense strategies in diverse bacterial species and communities.

Reviewer's comment: The authors clarified in the text of the revised manuscript that a selection of human commensal isolates was considered to be tested in vitro.

#30) How does the use of microbiota from a healthy individual contribute to our understanding of cholera, considering the unique microbiome in (often malnourished) people living in areas endemic for cholera. Please outline your rationale why you study a microbiome from a healthy person.

Authors' reply #30: This is again a very important point, which we very much appreciate. Notably, past volunteer studies and also animal models likewise do not reflect people in cholera endemic areas. Yet, healthy human volunteers can be infected – even though this requires an incredibly high infectious dose. These studies are commonly cited in the field, as they are in our manuscript. The underlying question of our study, which we developed in the manuscript, addressed the observations that on one hand *V. cholerae* is able to compete with human gut commensals, given that it can infect human adults, while, on the other hand, healthy people are highly refractory to infection. The rationale for the latter was therefore that the microbiota might play a role in defense in case *V. cholerae* would use its T6SS, as has been suggested (Zhao et al., 2018). More generally, studies on malnourishment and/or co-infections (for instance, with helminth) are indeed very rare or even non-existing in the cholera pathogenesis field. This is certainly something that should be addressed in the future. There is, however, some recent work on the microbiota of people in cholera endemic areas of the world (for instance, Chen et al., 2020 on environmental enteric dysfunction in Bangladeshi children but also Vonaesch et al., 2018 on stunted childhood growth in sub-Saharan African children). Unfortunately, these data cannot be used to support or refute our claims, as these studies used 16S sequencing data that do not provide any detailed insight into the commensals' genome content (e.g., presence/absence of T6SS cluster(s) in Enterobacter strains or capsule cluster(s) in *Klebsiella* species). We nonetheless discuss this topic in the revised manuscript and disclose the caveat of using samples from human volunteers.

Reviewer's comment: authors are claiming that microbiome from healthy people are considered in this study as authors are addressing that healthy people have a highly refractory effect to *V. cholerae* infection. Authors agree with the fact that previous studies were done with volunteers and also animal models that do not reflect cholera endemic areas. Authors explained the caveat in the revised manuscript.

#31) MINOR COMMENTS

Figure 1: Competitive index instead of survival should be shown (at least for *Klebsiella* and *Enterobacter cloacae*), which the authors show are resistant to *Vibrio cholerae*.

Authors' reply #31: We thank the reviewer for the comment but respectfully disagree. Indeed, the way we show the data is how it is commonly done in the field and in all our previous publications (see, for instance, Borgeaud et al., 2015, Science). Importantly, in Fig. 1 and several other figures, the values for both the prey and the predator are provided (input cell numbers are always adjusted to the same levels).

Reviewer's comment: Competitive index was suggested as it is more visual and easier to compare results. However, values for prey and predator are also valid.

#32) Figure 2a and 2b: $\Delta tssB1$, $\Delta tssB2$, $\Delta tssB1\Delta tssB2$ complementation are missing.

Authors' reply #32: We thank the reviewer for this comment. We did not consider these experiments as essential in the initial manuscript, as these specific *tssB* genes were situated in the middle of the large T6SS-1 clusters. For this reason, (polar) effects on neighboring genes would only affect other T6SS genes. It is correct, however, that we could not fully exclude that secondary mutation(s) might have occurred during strain construction in all those mutant strains and could therefore cause the killing deficiency. For this reason, we performed the requested complementation experiments and included them in the revised manuscript (Fig. S2).

Please note that, as explained at the beginning of this rebuttal letter, the initial *tssB* knockout mutants were indeed not complementable (most likely due to a polar effect). Consequently, we reconstructed new *tssB/tssB1* knockout strains. The outcome on interbacterial competition was, as expected, almost identical to the data provided for the initial *Enterobacter tssB/tssB1* mutants. For consistency we replaced all experiments that included any of the *tssB* mutant(s) with data using these new mutant(s), as explained in detail at the beginning of this rebuttal letter.

Reviewer's comment: The authors were unable to complement the *tssB* knockout mutants due to a polar effect in the initial manuscript. The authors successfully reconstructed *tssB/tssB1* knockout strains, changed the data sets of all experiments using *tssB* mutants with almost identical results. Moreover, the *tssB* mutant was successfully complemented in trans using a plasmid.

#33) Figure 2d and 2e: T6SS- controls are missing for strains other than #10 and #11, making it difficult to conclude it is T6SS-mediated killing? It seems that one T6SS- control corresponding to #10 is used as a control for the rest of the strains.

Authors' reply #33: We thank the reviewer for this comment. Indeed, in the initial manuscript, we did not delete the T6SS of all other killing-proficient strains, as the focus of our study was primarily on the commensal isolates. We now followed the reviewer's critique and tried to also delete the T6SS genes in the other killing-proficient *Enterobacter* strains for which genome sequences were available (e.g., DSM 30054, DSM 30060 and DSM 26481). Unfortunately, strain DSM 26481 was genetically intractable in our hands, despite several attempts containing many controls. However, we successfully modified strains DSM 30054 and DSM 30060 and the data fully support our claims (now presented in Fig. 2e and S5c).

Reviewer's comment: authors are claiming that strain DSM 26481 was intractable in their hands using allelic exchange. However, authors were able to get the T6SS knockout mutants on DSM 30054 and DSM 30060 and obtained results representing a non-functional T6SS. Please explain in the text why DSM 26481 could not be manipulated? Should this strain be pulled from this study?

#34) Figure 4c: NJST258-1 does not show protection against *V. cholerae*. The authors say that it could be due to the thickness of the capsule. This claim should be validated by microscopy.

Authors' reply #34: We thank the reviewer for this comment. Indeed, strain NJST258-1 is not entirely protected against T6SS assaults; however, there is still a ~10-fold protection compared to the unencapsulated mutant, which is statistically significant, as indicated in the figure. For this reason, we do not think that this finding contradicts our claim on CPS-dependent protection.

Unfortunately, the Indian Ink staining method is not meant to quantitatively measure capsule thickness. We therefore tried EM-based methods which were, however, unsuccessful due to Biosafety regulations that require extensive fixation of all samples before they can leave the dedicated BSL 2 laboratories. Collectively, we consider such thickness measurements outside the scope of the current study, as all commensal *Klebsiella* strains showed very striking and highly significant capsule dependent protection phenotypes.

Reviewer's comment: Again, the reviewers bring forward a regulatory argument before a scientific one. This is a trivial experiment to do.

Rebuttal letter by Flaugnatti, Isaac *et al.*

Once again, we thank the editor and the reviewers for the re-evaluation of our work.

- #1) Below, we provide a **point-by-point response to the individual comments**, as requested.
- #2) In general, we amended the text to delineate limitations and future directions, as requested by the editor.

We hope that the revised manuscript is now acceptable for publication.

Point-by-point response to the reviewers' comments.

Reviewer n'1 (Remarks to the Author):

#3) The authors have made a considerable effort to address the main issues raised by the three reviewers. As a result, the manuscript has been very much improved. Most of my experimental concerns have been properly addressed. I still disagree with some of the interpretations and with the way the work is presented...but this is not my paper! I believe this important work can be published as is, and the scientific community will take care of the rest. Therefore, I recommend acceptance.

Authors' reply #3: We thank the reviewer for their comment related to the improvement of the manuscript and that the experimental concerns have been properly addressed. As for the non-experimental aspects, we very much appreciate his/her decision to let us, the authors, decide how we present our data and the way we interpret the data. This is how peer review should work. Thank you!

Reviewer n'2 (Remarks to the Author):

#4) The authors provided careful and thoughtful responses to the reviewer comments and made many changes to the original manuscript that have strengthened their findings.

Authors' reply #4: We thank the reviewer for their kind words.

Reviewer n'3 (Remarks to the Author):

#5) SUMMARY

The authors replied to major and minor comments in detail and made changes in the revised manuscript. The authors also added two suggested key experiments to the revised manuscript: (1) knockout mutant in T6SS required as T6SS- controls. The authors successfully constructed the T6SS- controls in DSM 30054 and DSM 30060 but not in DSM 26481. (2) the authors complemented tssB/tssB1 knockout strains in trans with an episomal element. Unfortunately, infant mice infection, and EM-microscopy to measure capsule thickness were not considered. Although the authors agree that the suggested experiments would improve this study, they did not pursue them mostly for biosafety reasons. The authors state that the animal model has many limitations. While I agree with this statement, it should also be applied to nutrient agar plates exclusively used in this study. I suggest the author find collaborators who can conduct the animal studies. I think in-vivo studies are crucial (even as supplementary figures) for a publication. Other concerns remain untouched in this version and are listed below in italics.

Authors' reply #5: We thank the reviewer for this summary. There are indeed always additional experiments that could be done. The additional experiments suggested would indeed support the overall finding. However, as explained in the previous rebuttal letter, if these experiments were to fail, they wouldn't disprove our overall finding, as a plethora of parameters could be the reason for such failure. For this reason, they seem outside the scope of the current study, as also concluded by the editors.

Concerning “The authors state that the animal model has many limitations. While I agree with this statement, it should also be applied to nutrient agar plates exclusively used in this study.” => this had been addressed in the previous version of the revised manuscript, where we explicitly mentioned that the studies are based on *in vitro* work. Nonetheless, in this second revision we now explicitly state the limitation of this *in vitro* work and mention that future *in vivo* work is needed as a follow-up.

#6) – previously #28) MAJOR COMMENTS

OLD: The in-vitro data are clear cut. Commensals with an ability to fight back or protect themselves with a capsule are refractory to *V. cholerae*'s T6SS attack. But do 1x1 in-vitro killing assays properly mimic the complex space and communities of the gut? Sterile infant or adult mice need to be infected with their cocktail and determine how a change in T6SS+/capsulated commensals influences colonization of *V. cholerae*. This experiment could be done by barcoding each commensal and pathogen. If the hypothesis is correct, they should see a decrease in *V. cholerae* colonization as T6SS+/capsulated commensals increase.

Authors' reply #28: We very much appreciate this comment that the in-vitro data are clear cut; this was our intention. We also fully agree that in-vitro pairwise competition, as is currently state-of-the-art in T6SS research, is only a starting point to better understand complex bacterial communities. However, we consider the knowledge that commensal bacteria can resist T6SS assaults in immunity protein independent manners an important first step into this direction. The suggestion to colonize sterile infant or adult mice is certainly interesting but entirely outside the scope of this study for several reasons (including the lack of bioethical approval for animal experimentations, the requirement for the de novo setup of a mouse model, etc). Most importantly, the commensal isolates are derived from humans and might not do well in a heterologous host such as the mouse. Thus, while a successful experiment might provide support in favor of our hypotheses, a failed experiment would certainly not disprove our claims, given that there could be so many maladapted parameters in such an artificial setup (e.g., sterile = germ-free mice, or those treated with antibiotic; human commensals and not mice commensals; composition of the commensal cocktail and ratio between species; MOI of commensals and *V. cholerae*; timing; etc).

NEW: Reviewer's comment: Authors agree with the idea that the mice experiments is interesting. However, authors decided not to pursue this line of investigation (see Summary statement).

Authors' reply #6: Indeed, as stated above under #5, such experiments would be interesting but are clearly outside the scope of this study.

#7) – previously #29)

OLD: How well does the gut library represent a gut microbiome with obligate anaerobes in the majority? Are the authors looking at the tip of the iceberg in their competition assays? This might be an issue if the commensals tested depend on unculturable commensal bacteria to put up a commensal barrier. At the very least, the authors should discuss this in detail.

Authors' reply #29: This is a very interesting question. Indeed, common gut libraries might underrepresent obligate anaerobes for obvious reason (e.g., oxygen sensitivity for growth or survival). With respect to our study: as *V. cholerae* is known to colonize the small intestine, we do not necessarily expect that it would interact much/at all with obligate anaerobes. Instead, as we described in the text, we focused on Gram-negative Enterobacteriaceae, which are known to be present in the small intestine. Importantly, we chose this culture collection, as the authors of the published work provided genome sequencing data for the described commensals (Forster et al., 2019; Shao et al., 2019). Given that we primarily focused on those species that were resistant, it seems that they don't require additional commensal-derived signals to put up this barrier. For those species that are killed by *V. cholerae* (*E. coli*, *Hafnia*, etc), we can indeed not exclude that they might behave differently in vivo. For this reason, we now revised the text and added “under the tested conditions”. We also mentioned in the abstract that

we only tested a selection of human commensal isolates and that we did so in vitro. In a way our findings might indeed only be the tip of the iceberg and we expect a plethora of future studies showing similar defense strategies in diverse bacterial species and communities.

New: Reviewer's comment: The authors clarified in the text of the revised manuscript that a selection of human commensal isolates was considered to be tested in vitro.

Authors' reply #7: Indeed, this point had been addressed in the first round revision.

#8) – previously #30)

Old: How does the use of microbiota from a healthy individual contribute to our understanding of cholera, considering the unique microbiome in (often malnourished) people living in areas endemic for cholera. Please outline your rationale why you study a microbiome from a healthy person.

Authors' reply #30: This is again a very important point, which we very much appreciate. Notably, past volunteer studies and also animal models likewise do not reflect people in cholera endemic areas. Yet, healthy human volunteers can be infected – even though this requires an incredibly high infectious dose. These studies are commonly cited in the field, as they are in our manuscript. The underlying question of our study, which we developed in the manuscript, addressed the observations that on one hand *V. cholerae* is able to compete with human gut commensals, given that it can infect human adults, while, on the other hand, healthy people are highly refractory to infection. The rationale for the latter was therefore that the microbiota might play a role in defense in case *V. cholerae* would use its T6SS, as has been suggested (Zhao et al., 2018). More generally, studies on malnourishment and/or co-infections (for instance, with helminth) are indeed very rare or even non-existing in the cholera pathogenesis field. This is certainly something that should be addressed in the future. There is, however, some recent work on the microbiota of people in cholera endemic areas of the world (for instance, Chen et al., 2020 on environmental enteric dysfunction in Bangladeshi children but also Vonaesch et al., 2018 on stunted childhood growth in sub-Saharan African children). Unfortunately, these data cannot be used to support or refute our claims, as these studies used 16S sequencing data that do not provide any detailed insight into the commensals' genome content (e.g., presence/absence of T6SS cluster(s) in *Enterobacter* strains or capsule cluster(s) in *Klebsiella* species). We nonetheless discuss this topic in the revised manuscript and disclose the caveat of using samples from human volunteers.

New: Reviewer's comment: authors are claiming that microbiome from healthy people are considered in this study as authors are addressing that healthy people have a highly refractory effect to *V. cholerae* infection. Authors agree with the fact that previous studies were done with volunteers and also animal models that do not reflect cholera endemic areas. Authors explained the caveat in the revised manuscript.

Authors' reply #8: Indeed, this point had been addressed in the first round revision.

#9) – previously #31) MINOR COMMENTS

Old: Figure 1: Competitive index instead of survival should be shown (at least for *Klebsiella* and *Enterobacter cloacae*), which the authors show are resistant to *Vibrio cholerae*.

Authors' reply #31: We thank the reviewer for the comment but respectfully disagree. Indeed, the way we show the data is how it is commonly done in the field and in all our previous publications (see, for instance, Borgeaud et al., 2015, *Science*). Importantly, in Fig. 1 and several other figures, the values for both the prey and the predator are provided (input cell numbers are always adjusted to the same levels).

New: Reviewer's comment: Competitive index was suggested as it is more visual and easier to compare results. However, values for prey and predator are also valid.

Authors' reply #9: Indeed, this point had been addressed in the first round revision.

#10) – previously #32)

Old: Figure 2a and 2b: Δ tssB1, Δ tssB2, Δ tssB1 Δ tssB2 complementation are missing.

Authors' reply #32: We thank the reviewer for this comment. We did not consider these experiments as essential in the initial manuscript, as these specific tssB genes were situated in the middle of the large T6SS-1 clusters. For this reason, (polar) effects on neighboring genes would only affect other T6SS genes. It is correct, however, that we could not fully exclude that secondary mutation(s) might have occurred during strain construction in all those mutant strains and could therefore cause the killing deficiency. For this reason, we performed the requested complementation experiments and included them in the revised manuscript (Fig. S2).

Please note that, as explained at the beginning of this rebuttal letter, the initial tssB knockout mutants were indeed not complementable (most likely due to a polar effect). Consequently, we reconstructed new tssB/tssB1 knockout strains. The outcome on interbacterial competition was, as expected, almost identical to the data provided for the initial *Enterobacter* tssB/tssB1 mutants. For consistency we replaced all experiments that included any of the tssB mutant(s) with data using these new mutant(s), as explained in detail at the beginning of this rebuttal letter.

New: Reviewer's comment: The authors were unable to complement the tssB knockout mutants due to a polar effect in the initial manuscript. The authors successfully reconstructed tssB/tssB1 knockout strains, changed the data sets of all experiments using tssB mutants with almost identical results. Moreover, the tssB mutant was successfully complemented in trans using a plasmid.

Authors' reply #10: Indeed, this point had been addressed in the first round revision.

#11) – previously #33)

Old: Figure 2d and 2e: T6SS- controls are missing for strains other than #10 and #11, making it difficult to conclude it is T6SS-mediated killing? It seems that one T6SS- control corresponding to #10 is used as a control for the rest of the strains.

Authors' reply #33: We thank the reviewer for this comment. Indeed, in the initial manuscript, we did not delete the T6SS of all other killing-proficient strains, as the focus of our study was primarily on the commensal isolates. We now followed the reviewer's critique and tried to also delete the T6SS genes in the other killing-proficient *Enterobacter* strains for which genome sequences were available (e.g., DSM 30054, DSM 30060 and DSM 26481). Unfortunately, strain DSM 26481 was genetically intractable in our hands, despite several attempts containing many controls. However, we successfully modified strains DSM 30054 and DSM 30060 and the data fully support our claims (now presented in Fig. 2e and S5c).

New: Reviewer's comment: authors are claiming that strain DSM 26481 was intractable in their hands using allelic exchange. However, authors were able to get the T6SS knockout mutants on DSM 30054 and DSM 30060 and obtained results representing a non-functional T6SS. Please explain in the text why DSM 26481 could not be manipulated? Should this strain be pulled from this study?

Authors' reply #11: We are not "claiming" that this strain was not genetically tractable; it is a fact that it wasn't tractable in our hands, as mentioned in the manuscript (and this wasn't for a lack of try). Why this was the case is currently unclear and finding the exact reason seems unjustified here. Indeed, this doesn't come as a surprise to us at all; many bacterial strains are not easily manipulatable (even strains within the same species of model organisms, as we witnessed ourselves - even for *Vibrio cholerae* strains). For this reason, we truly believe that it is unnecessary to explain specifically for this strain why it is genetically intractable. We also don't think that pulling it from this study is of any help, as inclusion of this strain doesn't contradict the overall claim of our work.

#12) – previously ##34)

Old: Figure 4c: NJST258-1 does not show protection against *V. cholerae*. The authors say that it could be due to the thickness of the capsule. This claim should be validated by microscopy.

Authors' reply #34: We thank the reviewer for this comment. Indeed, strain NJST258-1 is not entirely protected against T6SS assaults; however, there is still a ~10-fold protection compared to the unencapsulated mutant, which is statistically significant, as indicated in the figure. For this reason, we do not think that this finding contradicts our claim on CPS-dependent protection.

Unfortunately, the Indian Ink staining method is not meant to quantitatively measure capsule thickness. We therefore tried EM-based methods which were, however, unsuccessful due to Biosafety regulations that require extensive fixation of all samples before they can leave the dedicated BSL 2 laboratories. Collectively, we consider such thickness measurements outside the scope of the current study, as all commensal *Klebsiella* strains showed very striking and highly significant capsule dependent protection phenotypes.

New: Reviewer's comment: Again, the reviewers bring forward a regulatory argument before a scientific one. This is a trivial experiment to do.

Authors' reply #12: We respectfully disagree. It's not solely a Biosafety issue. Indeed, as discussed in the manuscript, it might not be solely the thickness of the capsule but could also be the stiffness, compaction etc. of the capsular material. Measuring all of this is not trivial at all and clearly outside the scope of the current study.